

# The Modulation of Synoptic Weather Patterns and Human Activities on the Diurnal Cycle of Summertime Canopy Urban Heat Island in Yangtze River Delta Urban Agglomeration, China

Tao Shi[1], Yuanjian Yang[2,*], Lian Zong[2], Min Guo[2], Ping Qi[1], Simone Lolli[3]

[1]School of Mathematics and Computer Science, Tongling University, Tongling, 244000, China
[2]School of Atmospheric Physics, Nanjing University of Information Science and Technology, Nanjing, 210044, China
[3]CNR-IMAA, Contrada S. Loja, 85050 Tito Scalo (PZ), Italy

*Correspondence to*: Prof. Yuanjian Yang (yyj1985@nuist.edu.cn)

**Abstract.** The pronounced excess urban warming phenomenon during summer in the Yangtze River Delta Urban Agglomeration (YRDUA) has emerged as a significant challenge to the health and economy of urban residents amidst the accelerating urbanization process. Despite its undeniable importance, few studies have investigated the diurnal variability patterns of the canopy urban heat island (CUHI) in this region from the perspective of synoptic weather patterns (SWP) and human activities. This study integrated multiple source datasets, including meteorological station observations, high-resolution satellite imagery, and reanalysis data, to systematically analyze the diurnal patterns of the CUHII in YRDUA. Using objective classification and a machine learning model, we discovered notable diurnal patterns of CUHI intensity (CUHII), particularly higher levels of CUHII at night compared to day. Further analysis revealed that among six SWPs, type 2, dominated by subtropical high pressure, generated the strongest CUHII, while type 4, influenced by the combined effects of southwestern moisture transport and southward cold air incursions, resulted in the lowest CUHII. Additionally, the study found that key indicators such as landscape percentage (PLAND), largest patch index (LPI) and anthropogenic heat flux (AHF) exhibited an increasing trend over recent years, with higher values in the east and lower in the west, aligning well with the spatio-temporal patterns of the CUHII. These findings collectively confirmed the central roles of SWPs and human activities as the main drivers of CUHI phenomena. Simulations using the RF model further indicated the diurnal asymmetry in the modulation of the CUHI by SWPs and human activities: SWPs exerted a more pronounced influence on day CUHII, while human activities dominated night CUHII. Furthermore, the study delved into the impact mechanisms of heatwave (HW) events on the diurnal cycle of the CUHII. During HW periods, the amplification effect of day CUHII was stronger than night CUHII, and the presence of HW events significantly reduced the diurnal amplitude of the CUHII. In conclusion, this study not only provided scientific insight into the complex driving mechanisms of the CUHI diurnal cycle in YRDUA, but also offered a theoretical foundation for evaluating urban overheating issues and developing effective mitigation strategies.



## 1 Introduction

The expansion of urban areas and the unprecedented growth of the population have led to the well-known phenomenon of the urban heat island (UHI) (Roth, 2007; Rizwan et al., 2008; Oke et al., 2017). Among the various manifestations of UHI, the CUHI refers to the temperature differential observed between urban and rural areas, specifically spanning the range from

the ground surface to the roof of urban structures (Liu et al., 2007; Yang et al., 2023). This phenomenon is closely linked to human welfare, as it exerts direct and indirect impacts on human comfort and health, energy consumption patterns, and even financial losses (Muthers et al., 2017; Salimi & Al-Ghamdi, 2020; Xia et al., 2018; Herbel et al., 2018; Marks & Connell, 2023; Singh et al., 2023; Yang et al., 2023).). YRDUA is one of the most developed, densely populated and concentrated industrial areas in China. In YRDUA, cities and regions are closely linked with each other. The diversity and spatial

heterogeneity of the land surface conditions, the dense population, and the close interconnection of the city regions make the YRDUA an ideal area for the study of the CUHI city agglomeration (Dong et al., 2014; Du et al., 2016; Zhang et al., 2022; Yan & Zhou, 2023).).

The CUHI phenomenon exhibits pronounced temporal variability throughout the diurnal cycle (Liu et al., 2022; Bansal & Quan, 2024; Lin et al., 2024). Specifically, it tends to intensify significantly after sunset, reaching its peak, while during the

day, its influence is notably weaker (Tong et al., 2018; Zhang et al., 2022). In fact, the intensities and causes of the day and night CUHI differ. The diurnal CUHI is usually caused by excess heat dissipated from urban surfaces through turbulent transfer, while the night CUHI is primarily caused by the heat stored in urban surfaces during the day (Giridharan et al., 2004, 2005). day urban excess warm events have the potential to induce heatstroke and exacerbate ground-level ozone pollution (Filleul et al., 2006; Gosling et al., 2009; Pu et al., 2017). However, urban excess warm events at night can hinder the body's

ability to recover during sleep, potentially leading to insomnia and abnormal temperature regulation (Le Tertre et al., 2006; Gosling et al., 2009; Fischer & Schär, 2010). Furthermore, research has also revealed that excess warm events at night in urban settings can negatively impact agricultural productivity, leading to reduced crop yields (Bahuguna et al., 2017). In the context of global warming and rapid urbanization, exploring the diurnal cycle of CUHI is of significant importance for understanding the impacts of excess urban warming on human health and social activities.

In recent years, scholars have conducted extensive research on the driving mechanisms of CUHI phenomena (Li et al., 2020; Jiang et al., 2019; Imran et al., 2019). Anthropogenic activities (for example, changes in land use / land cover, anthropogenic heat, and aerosols) can also modulate urban excess warming events (Ren et al., 2015; Zheng et al., 2020). Currently, numerous scholars have only used land use and land cover data to investigate the local urban thermal environment (Ren & Ren, 2011; Shi et al., 2015; Tysa et al., 2019; Xue et al., 2023; Shi et al., 2024). However, the urban thermal effect is not

only influenced by the properties and scales of the underlying surfaces; the landscape pattern also plays a pivotal role that cannot be overlooked (Ren et al., 2015; Estoque et al., 2017; Chen et al., 2022). Specifically, when the built-up area remains constant, the air temperature increases in tandem with the increase in the building patch index (Shi et al., 2021). Anthropogenic heat, which encompasses heat sources such as buildings, transportation, and industrial emissions,





significantly contributes to the urban thermal environment (Guo et al., 2021). Previous studies have shown that aerosol particles intensify the CUHI phenomenon (Menon, 2002; Poupkou et al., 2011; Zheng et al., 2018), but some studies also demonstrated the opposite effects of particulate matter (Yang et al., 2020; Wu et al., 2021). Overall, the driving mechanism of human activities on the CUHI phenomenon is still under continuous exploration.

Certain synoptic weather patterns (SWPs) can cause noticeable changes in CUHI through their modulation of boundary layer meteorological factors (Hoffmann & Schlünzen, 2013; IPCC, 2021; Yang et al., 2022; Zhang et al., 2024). The subtropical high of the western Pacific is an important factor in the monsoon system, which generates high temperatures in southeastern China (Wang et al., 2015). There is significant interannual variability in the extent, intensity, and location of the WPSH, and its positional configuration with the westerly jet and South Asian high affects the region where high temperatures occur. Under clear and cloudless conditions, the solar shortwave radiation received by the ground surface intensifies during the day (Hong et al., 2018), while light winds further mitigate the horizontal dispersion of near-surface heat (Tong et al., 2012), thus rendering local high-temperature events more stable and persistent. High-pressure systems in summer can suppress the development of the planetary boundary layer and induce calm and cloud-free conditions favorable for radiation enhancement, thus raising temperatures (Miao et al., 2017; Yang et al., 2018; Wang et al., 2017a).). However, to date, there remains a significant knowledge gap in understanding how SWPs and human activities modulate the diurnal cycle of the CUHI in YRDUA. Specifically, how do we quantitatively assess the relative importance of SWPs and human activities on the diurnal cycle of the CUHI? Additionally, do they exist distinct driving effects on daytime CUHI and nighttime CUHI?

Addressing the aforementioned questions, this study used multisource meteorological and environmental data to objectively classify SWPs in YRDUA. Subsequently, a machine learning method was employed to explore the modulation mechanisms of both SWPs and human activities in the diurnal cycle of the CUHI. The overarching goal of this research was to provide valuable insights into the mitigation of urban overheating and the management of urban planning, thus fostering a deeper understanding of the intricate interaction between natural factors and human factors in the shaping of the urban thermal environment.

## 2 Data and methodology

### 2.1 Study Area

YRDUA is considered one of the influential world-class metropolitan regions, playing a pivotal role in China's economic and social development processes (Tian et al., 2011). As shown in Figure 1a, YRDUA is situated in the mid to lower reaches of the Yangtze River, serving as the junction between the eastern coastal region and the Yangtze River basin in China. YRDUA denotes the administrative region comprising Shanghai, Jiangsu, Zhejiang, and Anhui (Fig. 1b). It represents one of the most developed, densely populated, and highly concentrated industrial zones in China. With a total area of 358,000 square kilometers, accounting for less than 4% of the country's total, the region is home to approximately 236 million people, inhabited by around 17% of the national population. In 2023, Shanghai's GDP reached 0.67 trillion USD, Jiangsu's GDP



amounted to 1.82 trillion USD, Zhejiang's GDP totaled 1.17 trillion USD, and Anhui's GDP stood at 0.67 trillion USD; collectively, these statistical data surpassed 4.26 trillion USD, accounting for one-quarter of China's total economic output. However, this rapid urbanization has led to a series of intricate urban environmental issues, with the CUHI phenomenon being particularly prominent (Huang et al., 2015; Du et al., 2016; Zhang et al., 2022).

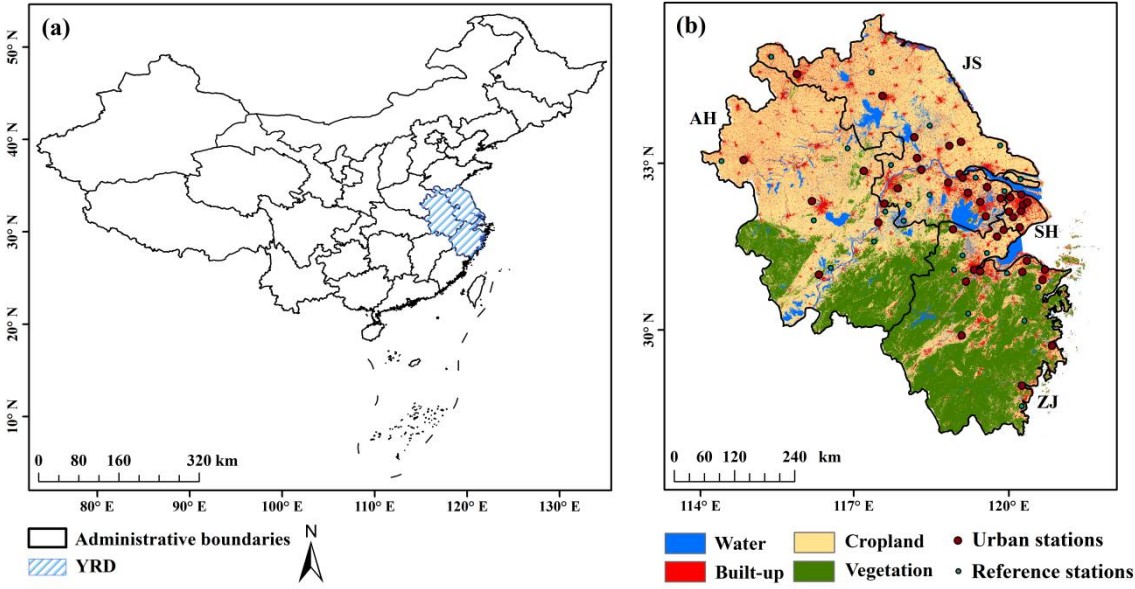

**Figure 1: (a) Overview of the study area. (b) Distribution of urban stations and reference stations in YRDUA.**

## 2.2 Data

### 2.2.1 Reanalysis data

The reanalysis data used in this study, comprising 850 hPa geopotential height, wind speed (WS), total cloud cover (TCC), relative humidity (RH), boundary layer height (BLH) and vertical velocity, were derived from the fifth generation European Center for Medium-Range Weather Forecasts (ECMWF) Reanalysis of Global Climate and Weather (ERA5).). This data set boasts a temporal resolution of hourly and a spatial resolution of 0.25°×0.25°. The data set was created by the Copernicus Climate Change Service, operated by ECMWF, and ERA5 data can be downloaded from https://cds.climate.copernicus.eu/cdsapp#!/home.

### 2.2.2 Observation data

In this paper, we used the hourly temperature, WS, and RH data covering the YRDUA provided by the China Meteorological Data Service Center (http://Data.cma.cn/en), spanning the months of June to August from 2011 to 2020. Daily, monthly and annual data used in this study were derived from hourly measurements. To uphold the integrity and rigor of the dataset, we





implemented a quality control procedure following the methodologies outlined by Xu et al. (2013) and Yang et al. (2011). Specifically, missing values within the observational sequences were substituted with the mean values of synchronous observations from the five nearest neighboring stations surrounding the target station. Stations with an excessive number of error records were excluded from the analysis. BLH data from the sounding stations were calculated on the basis of the

methodologies described by Seidel et al. (2012) and Guo et al. (2019). The concentration data of PM2.5 and PM10 could be accessed from the following links: https://doi.org/10.5281/zenodo.5652265 (Bai et al., 2021a) and https://doi.org/10.5281/zenodo.5652263 (Bai et al., 2021b), respectively.

### 2.2.3 Remote sensing data

The anthropogenic heat flux (AHF) data were derived from the inversion of the National Oceanic and Atmospheric Administration (NOAA) night light satellite dataset (http://ngdc.NOAA.gov/eog/dmsp/downloadV4composites.html), with a calculation error margin of less than 12% (Chen et al., 2016).

The annual China Land Cover Dataset (CLCD) is a dynamic dataset of land use released by Wuhan University. Yang & Huang (2021a) developed the land cover datasets with a spatial resolution of 30 m based on 335,709 Landsat images on the

Google Earth Engine platform. The latest dataset contains information on land cover for China from 1985 to 2021, and the overall precision of land classification is 80%.

The normalized difference vegetation index (NDVI) data set used in this study, produced and distributed by the National Ecological Science Data Center (http://www.nesdc.org.cn/), has a spatial resolution of 30 meters and a temporal resolution of one year. It involves the removal of clouds and shadows by obtaining all valid Landsat observations, followed by the

calculation of the NDVI index for each Landsat observation. Subsequently, through a combination of interpolation and smoothing techniques, the maximum NDVI value is obtained for each pixel location throughout the year (Yang et al., 2019).

In this paper, a buffer zone with a radius of 5km centered around each station was defined as the calculation area, from which various human activity factors could be obtained for each station, including the percentage of landscape (PLAND), the largest patch index (LPI), NDVI, and AHF.


### 2.3 Methods

### 2.3.1 Stations selection and CUHII calculation

Since surface air temperature is measured at a height of 2 meters, previous studies (Yang et al., 2013; Cai, 2008; Shi et al., 2015) have indicated that under conditions of advection and turbulent transport, the maximum impact of anthropogenic heat

on meteorological observations within a station typically does not exceed 5 km. Consequently, a radius of 5 km was selected as a buffer zone to capture the effects of urbanization on air temperature. AHF serves as an indicator of the influence of human emissions and changes in land use on sensible and latent heat fluxes in the lower atmosphere (Jiang et al., 2019; Chen et al., 2020). Following the calculation of the average AHF within a 5-km radius around each station, the top a third of the stations, ranked by their AHF values, were designated as urban stations (USs) for this study.



The selection of reference stations (RSs) is the key step in calculating the CUHII (Ren & Ren, 2011). The stations in the bottom one-third, ranked by their AHF values, were chosen as candidate RSs. Furthermore, RSs must meet the following criteria: they must have continuous records spanning over 50 years without missing data; the number of relocations must be less than three, and any relocations must involve a horizontal distance of less than 5 km (Zhang et al., 2010; Ren et al., 2015; Shi et al., 2015; Wen et al., 2019; Yang et al., 2022).). As a result, 43 USs and 27 RSs were selected for this analysis. By calculating the temperature difference between USs and RSs, the summer CUHII in YRDUA was ultimately derived (Ren et al., 2007; Yang et al., 2022).

### 2.3.2 Synoptic weather classification

The T-mode principal component analysis (T-PCA) method is an objective classification method. Initially, T-PCA standardized the weather data spatially and divided them into 10 subsets. Subsequently, the principal components (PC) of the weather information were estimated through singular value decomposition, and the corresponding PC scores were calculated after oblique rotation. Finally, the resultant subset with the highest sum was selected by comparing the 10 subsets based on contingency tables, and the classification result for this subset could be output (Miao et al., 2017; Philipp et al., 2014). This paper objectively classified synoptic circulations during the summer period of 2011 to 2020, focusing on the field of geopotential height of 850 hPa within the geographical range of 0 ° 60 ° N and 60 ° 150 ° E. Six summer weather patterns (SWPs) for YRDUA were identified (as shown in Fig. 2). The frequency of each SWP type in each month was defined as the number of days of occurrence divided by the total number of days.

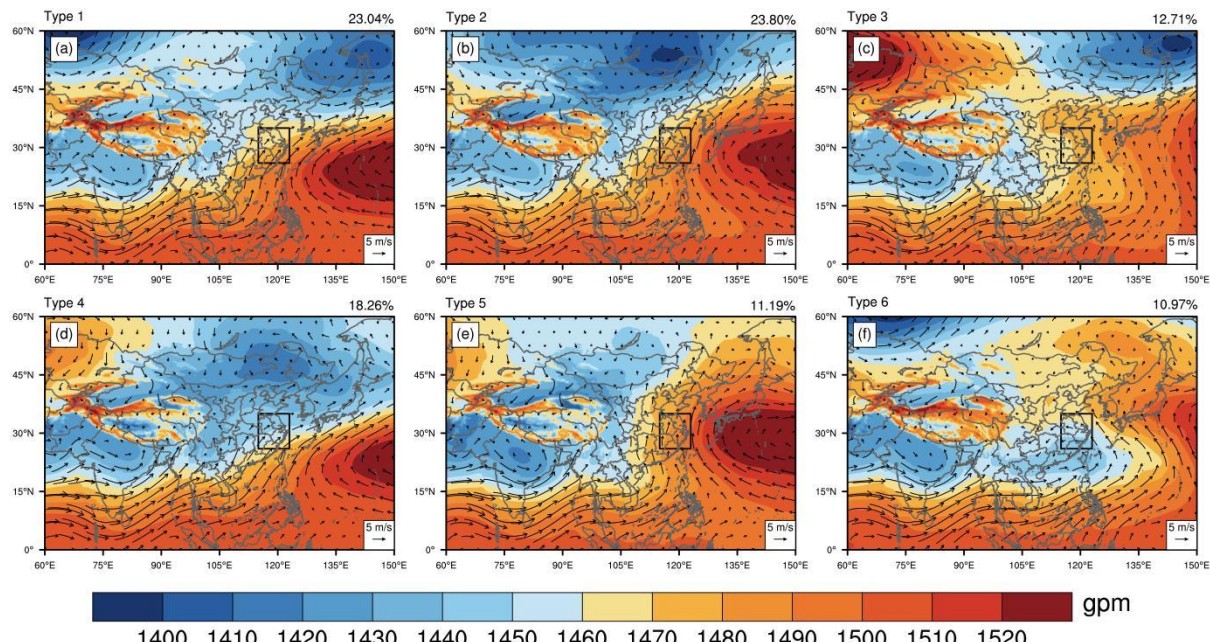

**Figure 2: The geopotential height of 850 hPa (shading) and wind (vectors) based on the objective classification in six SWPs in summer (a~f), respectively. The black box indicating the YRDUA.**





Under type 1 (23.04% of occurrences), abundant moisture transport from the southwestern sea region and prevailing southwest winds create favorable conditions for water vapor, conducive to precipitation formation. In type 2 (23.80%), the subtropical high jumps northward, placing the YRDUA under its control, with decreased moisture transport from the

southwestern sea, resulting in the highest frequency of occurrence among the six types. Type 3 (12.71%) features the subtropical high retreating eastward, accompanied by low WS. Type 4 (18.26%) involves the subtropical high retreating southward and eastward, influenced by both moisture transport from the southwestern sea and southward-moving cold air, favoring precipitation and temperature reduction. In type 5 (11.19%), YRDUA is primarily controlled by subtropical high, with warm air transported from the southeastern ocean promoting air subsidence, which is conducive to high-temperature

weather. Lastly, type 6 (10.97%) sees a small cyclone center emerging in central and southern China, positioning the YRDUA in the vicinity of a weak low-pressure system, potentially influencing local weather patterns.

### 2.3.3 Random forest model

The Random Forest (RF) model, an extension or evolution of decision trees, represents a popular and highly versatile

machine learning approach (Tan et al., 2017; Yu et al., 2020). Unlike traditional linear regression models, RF operates as a nonparametric method, capable of modeling complex nonlinear relationships among predicted values and various predictor variables (Hastie et al., 2009), while also identifying the significance of individual variables (Wang et al., 2019).). Based on previous research (Duan et al., 2021; Chen et al., 2022), we randomly divided the stations within the YRDUA into train (70%) and test (30%) samples. With CUHII serving as the dependent variable, the RF model incorporated both synoptic

factors and anthropogenic factors as independent variables, encompassing SWPs, RH, WS, BLH, AL, PLAND, LPI, AHF, PM2.5, and PM10. To train and test the RF model, we employed a 10-fold cross-validation approach (Zeng et al., 2020). The construction of the RF model and the calculation of importance scores for influencing factors were implemented using Python.

## 3 Results

### 3.1 Diurnal cycle of the CUHII in YRDUA

On the background of climate warming, human activities have produced considerable amounts of anthropogenic heat and pollutant emissions, which, to a certain extent, exacerbate urban excess warming.



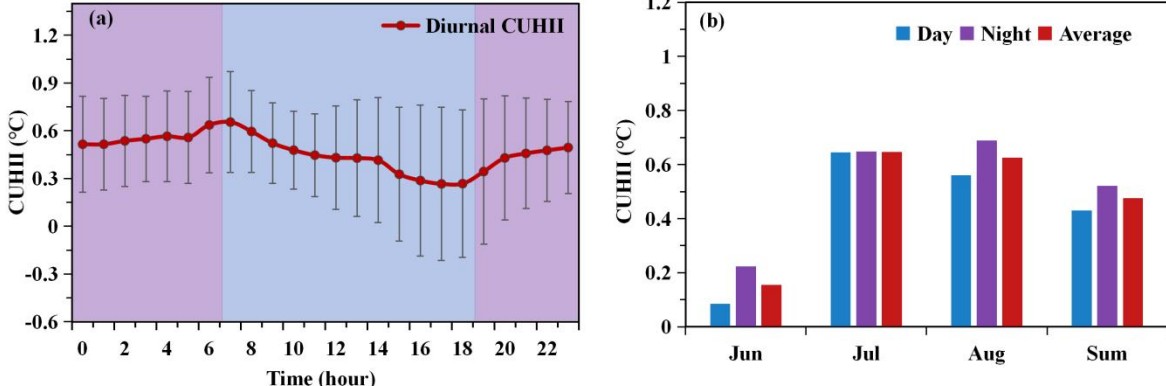

**Figure 3: Temporal characteristics of CUHII during day and night. (a) The diurnal variation of CUHII, with short lines indicating**
**standard deviation, blue areas representing day, and purple areas representing night. (b) The monthly variation of CUHII during**
**day and night.**

Fig. 3a illustrates the hourly variation of CUHII in the summer in YRDUA from 2011 to 2020. At 8:00 Beijing time (BJ,
same below), as the solar altitude angle increases, the temperature in suburban areas rises faster than that in urban areas.
Coupled with higher WS during the day compared to night, turbulence intensifies, leading to a rapid decline in the urban-
suburban temperature difference. Consequently, CUHII reaches its minimum value of 0.27 ° C at 17:00 BJ. After 18:00 BJ,
as the solar altitude angle decreases, the effective radiation in suburban areas gradually increases, accelerating atmospheric
heat loss. Since urban areas accumulate more heat, long-wave radiation from the ground continues to supply heat to the
atmosphere, resulting in a rapid widening of the urban-suburban temperature difference. Before sunrise, between 0:00 and
7:00, the cooling rates of urban and suburban temperatures are similar, causing the CUHII to gradually increase to its daily
maximum value of 0.65 ° C. Overall, the CUHII exhibits a clear diurnal cycle characterized by a gradual decrease, stable low
values, rapid increase and stable high values, with pronounced day-night differences, consistent with previous studies (Wang
et al., 2017b; Zhang et al., 2022).). Fig. 3b depicts the intraseasonal variation of CUHII in YRDUA. It can be observed that
the average CUHII in July and August (approximately 0.63°C) is significantly higher than in June, and the day-night
difference in CUHII in June and August (approximately 0.14 ° C) is significantly greater than in July. Throughout the entire
summer period, the average CUHII during night is 21.11% higher than during day.



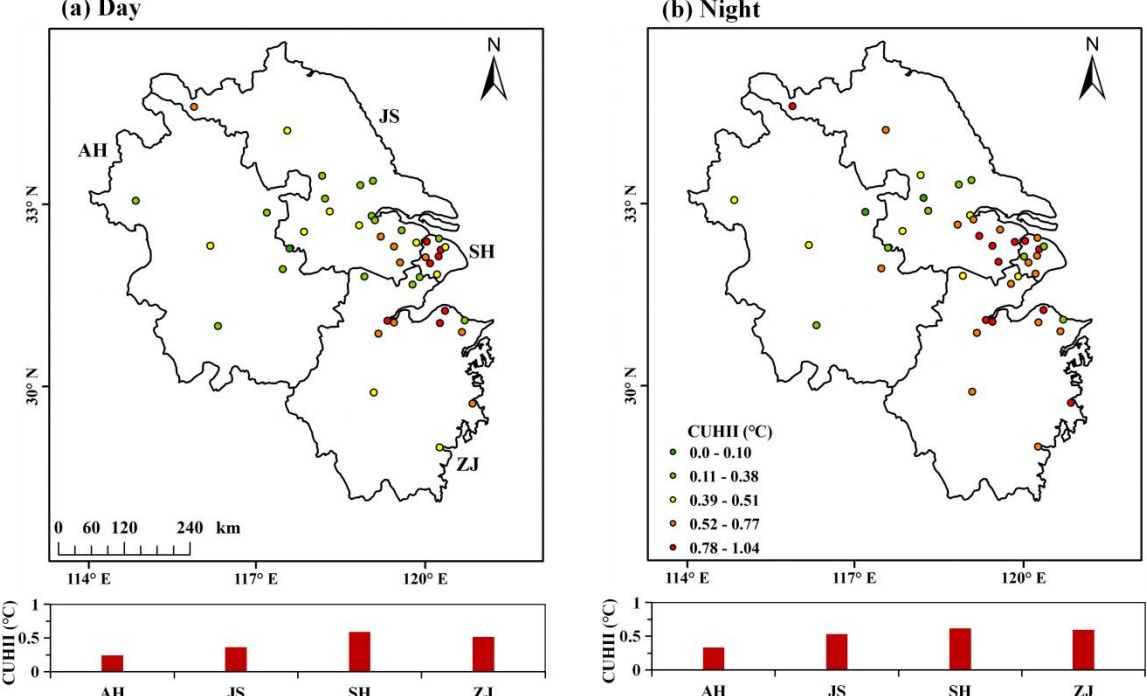

**Figure 4: Spatial patterns of CUHII in YRDUA during the day (a) and the night (b). Different colored dots represent different ranks of the CUHII. The bar chart below represents the average CUHII of Anhui (AH), Jiangsu (JS), Shanghai (SH) and Zhejiang (ZJ).**

Fig. 4 illustrates the spatial patterns of CUHII in YRDUA. Taking the day CUHII as an example (Fig. 4a), it is evident that the CUHII in the eastern YRDUA is significantly higher than that in the western YRDUA. Specifically, SH exhibits the highest CUHII, reaching 0.59°C, with the highest CUHII observed at the Xujiahui station, peaking at 0.95 ° C. Following SH, ZJ and JS rank second and third, with CUHII values of 0.52 ° C and 0.37°C, respectively. In contrast, AH has the lowest CUHII, at merely 0.25 ° C. When considering the night CUHII (Fig. 4b), SH maintains the highest CUHII, which rises to 0.62°C, while the CUHII at the Xujiahui station increases even further, reaching 1.04 ° C. Similarly, ZJ, JS, and AH also experience varying degrees of intensification of CUHII. In conclusion, there is a pronounced difference in the spatial-temporal patterns of CUHII between day and night. The underlying mechanisms driving this phenomenon will be analyzed from the perspectives of SWPs and human activities in the subsequent sections of this paper.

**3.2 Spatial-temporal patterns of SWPs and human activities in YRDUA**

Based on the T-PCA results, the summer synoptic backgrounds in YRDUA from 2011 to 2020 can be classified into six distinct SWPs. We first conducted a statistical analysis of the occurrence frequencies of these different SWPs, as illustrated in Fig. 5.



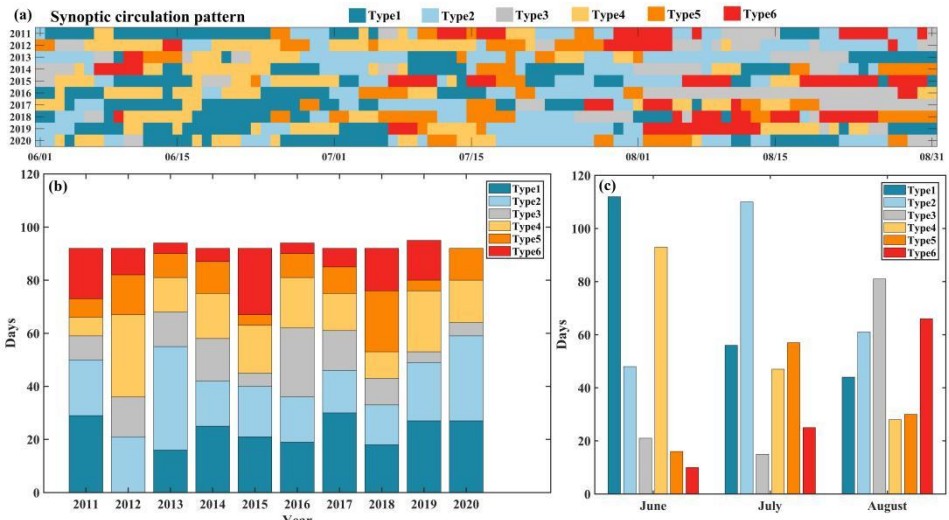

**Figure 5: (a) Daily, (b) Interannual and (c) Monthly occurrence frequencies of the six SWPs in YRDUA from 2011 to 2020.**

The daily, interannual and monthly frequency of occurrence of six SWPs during the summer from 2011 to 2020 revealed

pronounced variations in atmospheric circulation patterns. Specifically, type 1 predominantly occurred in late June and early July 2011, as well as in 2017-2020. Type 2 exhibited the highest frequency, mainly concentrated in mid-to-late July and early August of each year, with particularly high occurrences in 2013 and 2020. Type 3 appeared frequently in August annually. Type 4 was more prevalent in mid to late June of each year. Type 5 was mostly observed in mid-July. Lastly, type 6 predominantly emerged in early August 2011-2012 and 2018-2019.

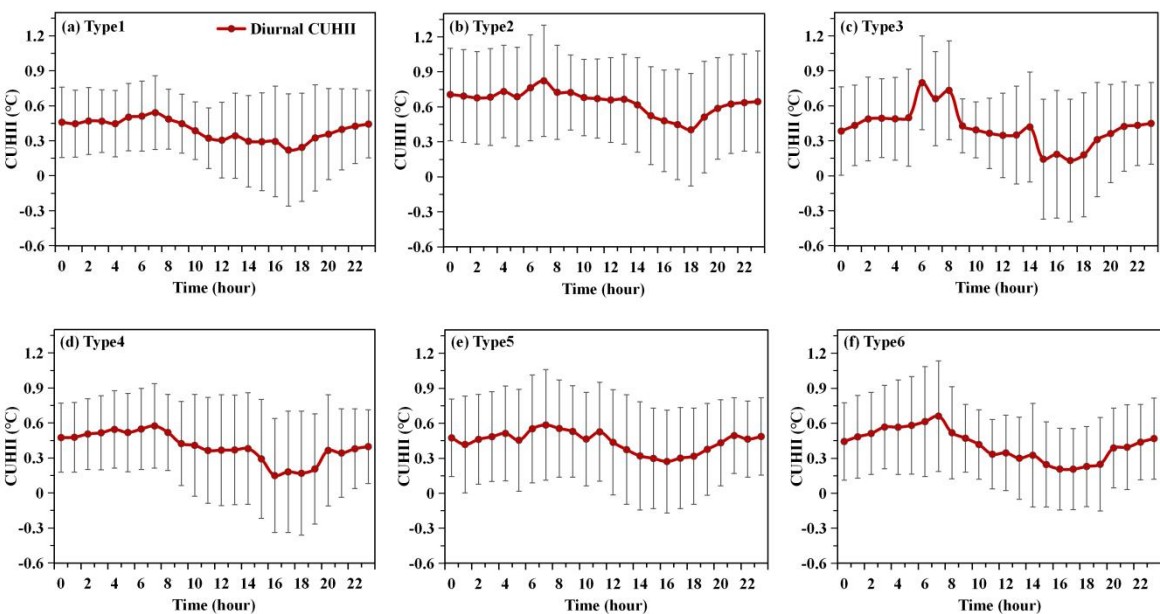




**Figure 6: Diurnal variations of CUHII in YRDUA under different SWPs (a-f).**

Next, we delved into analyzing the diurnal variation of CUHII in YRDUA under different SWPs. As evident from Fig. 6, under all SWPs, the diurnal variation of the CUHII consistently exhibits a periodic pattern of gradual decline - stable low values - rapid increase - stable high values. Specifically, the daily maximum of the CUHII occurs under type 2, reaching 0.82 ° C. Under type 2, with the northward movement of the subtropical high of the western Pacific, the Meiyu season ends, transitioning into a period dominated by hot and dry weather, characteristic of midsummer. On the contrary, the daily minimum of the CUHII is observed under type 4, at merely 0.14 ° C. Under type 4, intensified low-pressure activities and moisture transport lead to cloudy skies, bringing about a precipitation-dominated climatic phase in YRDUA. Fig. 7 illustrates the spatial patterns of the CUHII under various synoptic backgrounds. In general, all high-CUHI centers align well with economically developed and densely populated urban areas of all types. Regionally, the CUHII in the eastern YRD is significantly higher than in the western YRD. SH records the highest CUHII, followed by ZJ and JS, while AH has the lowest. In terms of SWP, the average CUHII under type 2 is markedly higher than that of other types, with day and night values of 0.65 ° C and 0.71°C, respectively. On the contrary, type 4 exhibits the lowest average CUHII, with day and night values of 0.41 ° C and 0.47°C, respectively. These findings underscore the crucial role of various SWPs in modulating CUHII in YRDUA.



**Figure 7: Spatial patterns of CUHII in YRDUA during the day (a) and night (b) under different SWPs. The bar chart on the right represents the average CUHII under different SWPs.**




The meteorological conditions of the boundary layer constitute important factors influencing the spatio-temporal variations of the CUHII (Ren et al., 2007; Yang et al., 2019; Yang et al., 2023). We delved into analyzing the spatiotemporal patterns of various local meteorological elements. Taking WS as an illustrative case, as the sun rises, the ground warms up, causing atmospheric stratification to become unstable and enhancing turbulence, which subsequently leads to a rapid increase in WS.

This trend is diametrically opposed to the diurnal variation of CUHII. The daily maximum WS is reached at 15:00, followed by a gradual decline (Fig. S1). In particular, day WS is markedly higher than night WS. Fig. S2 reveals that type 6 exhibits the highest WS, attributable to the highest boundary layer observed over the YRDUA under type 6 conditions, where a weak low-pressure system contributes to the upward development of the boundary layer. On the contrary, type 4 displays the lowest WS, which corresponds to the lowest boundary layer and increased cloud cover, thereby reducing solar radiation

reaching the ground and inhibiting boundary layer growth. Figs. S3 and S4 indicate that as air temperature rises after sunrise, the saturation vapor pressure increases, leading to a decrease in RH, which reaches its daily minimum at 15:00 before gradually rising thereafter. The day humidity is conspicuously lower than the night humidity. Among all types, type 1 exhibits the highest humidity, while type 2 shows the lowest, though with minimal differences between the two. Turning our attention to the BLH, during night, the atmospheric stratification is stable, accompanied by low WS, which consequently

results in a low BLH. However, as the sun rises and wind speeds intensify, the BLH begins an upward trend (see Fig. S5).





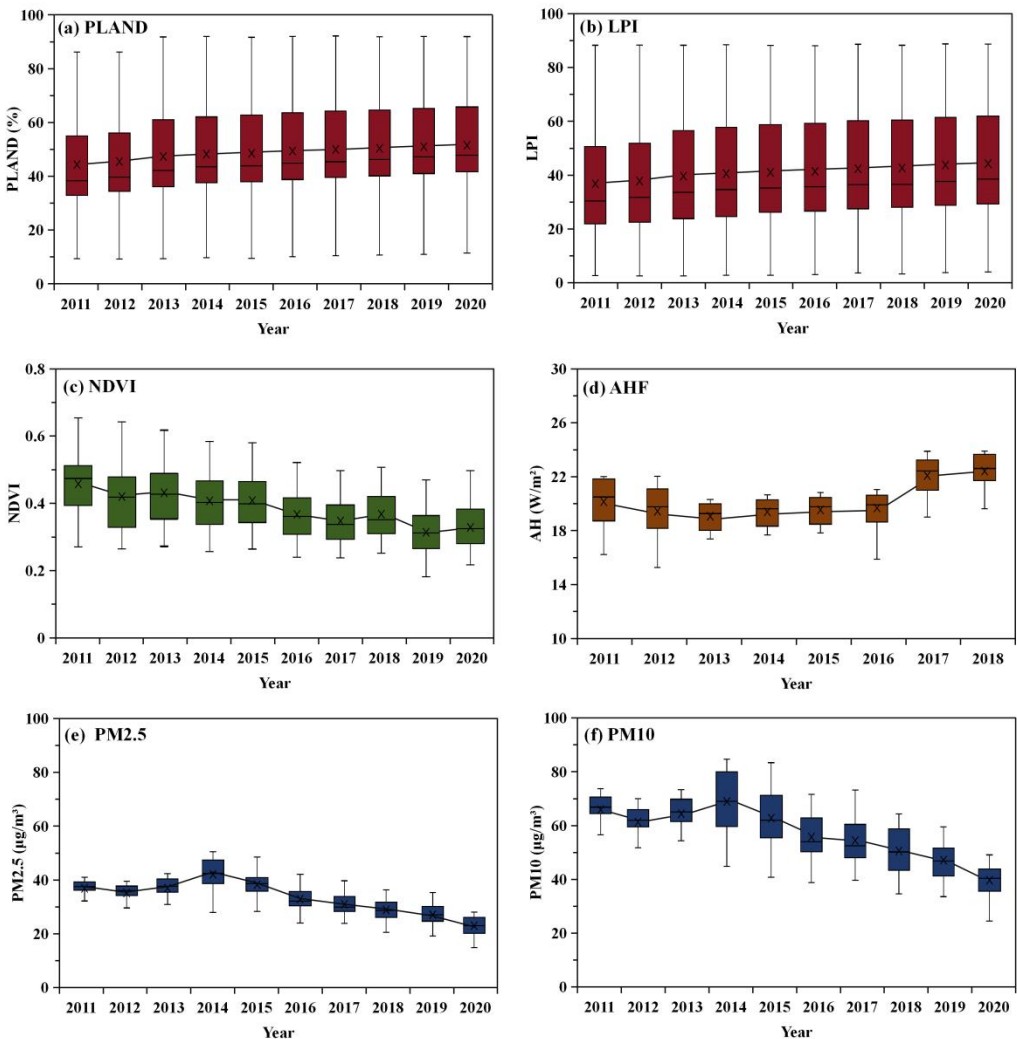

**Figure 8: Temporal patterns of PLAND (a), LPI (b), NDVI (c), AHF (d), PM2.5 (e), and PM10 (f) in YRDUA.**

Previous studies have demonstrated that human activities, such as land use, anthropogenic heat, and aerosols, are significant
drivers of the CUHI phenomenon (Ren et al., 2015; Zheng et al., 2020; Yang et al., 2023). As urbanization progresses, the
YRDUA, one of the most economically vibrant and developed regions in China, has seen a notable and sustained expansion
of its built-up areas. Figs. 8a and 8b illustrate that PLAND gradually increased from 44.31% in 2011 to 51.91% in 2020, and
LPI also rose from 37.01 in 2011 to 44.65 in 2020. The spatial patterns of PLAND and LPI (Fig. 9a and Fig. 9b) indicate that
SH exhibits the highest level of urbanization, as evidenced by half of the stations appearing in a deep red grade on the map,
followed by ZJ and JS, and AH showing the lowest level, which generally corresponds to the spatial patterns of the CUHII.
Fig. 8c reveals a declining trend in NDVI over the years, and the spatial pattern of NDVI exhibits an opposite pattern to that



of the CUHII (Fig. 9c). Since AHF is closely related to changes in the built-up areas surrounding the meteorological stations (Guo et al., 2021), the temporal-spatial patterns of AHF are generally consistent with those of PLAND (Fig. 9d). Furthermore, PM2.5 and PM10 concentrations exhibit a trend of initial growth followed by a decrease (Fig. 8e and Fig. 8f),

suggesting an improvement in air quality after an initial period of deterioration. Following the issuance of the "Action Plan for Air Pollution Prevention and Control" released by the State Council in 2013, various pollution prevention and control measures have been implemented across the YRDUA, leading to a marked improvement in air quality. The gradual decrease in aerosol concentrations from northwest to southeast within the YRDUA (Fig. 9e and Fig. 9f) may be attributed to differences in industrial structures, infrastructure, and environmental protection policies between different regions (He et al.,

300 2024).).

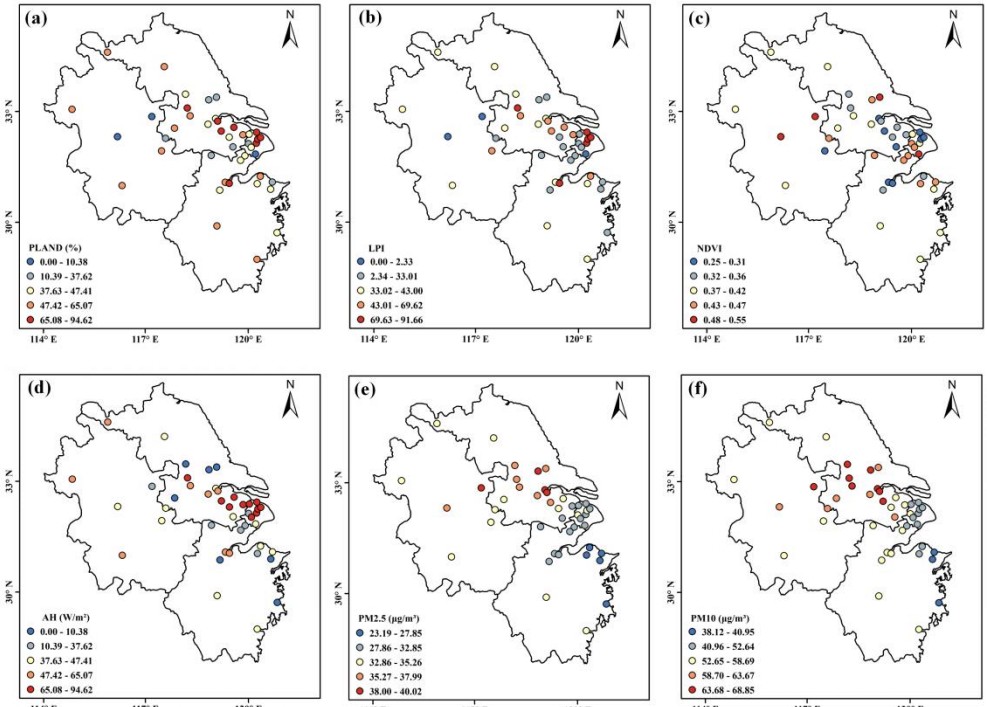

**Figure 9: Spatial patterns of PLAND (a), LPI (b), NDVI (c), AHF (d), PM2.5 (e) and PM10 (f) in YRDUA from 2011 to 2020.**

**3.3 The modulation of the CUHII by SWPs and human activities**

In this section, we selected synoptic backgrounds, meteorological conditions, and urban morphology as influencing factors

and used the RF model to fit the day CUHII and night CUHII, aiming to explore the driving mechanisms of synoptic and human factors on the CUHII. Fig. 10 compares the performance of the RF models for day and night. During the day (Fig. 10a), the RF model achieves an R-squared (R2) value of 0.95 and a root mean squared error (RMSE) of 0.13oC in the train data, indicating an excellent fit between the model predictions and the observed data. When we turn our attention to the test





data, the performance of the RF model decreases, which might be attributed to differences in the distribution between the test

and train data. Similarly, for night (Fig. 10b), the RF model produces excellent results on the train data. These findings

suggest that the RF model might be a powerful tool for simulating the local urban thermal environment (Yu et al., 2020;

Chen et al., 2022).

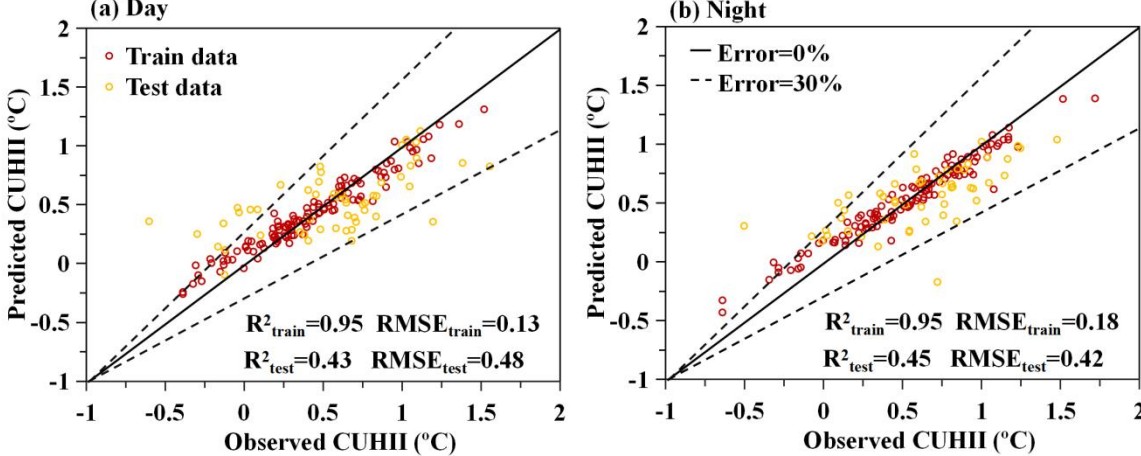

**Figure 10: Prediction results of day CUHII (a) and night CUHII. Red circles represent the train data, while yellow circles**
**represent the test data. The dashed lines indicate 30% fit error lines.**

Next, we used the RF model to analyze the contributions of various factors to day CUHII and night CUHII. Shapley additive

explanation (SHAP) is an interpretability framework used to explain model outputs (Park et al., 2023). Assesses the impact

of individual characteristics on prediction results by quantifying their contributions to the result. As depicted in the left

subplot (SHAP value), each row represents a feature, with the horizontal axis indicating the SHAP value. Each dot

corresponds to a sample, where the reddish colors signify higher feature values, and the bluer colors indicate lower values. It

is evident that RH exerts the most significant influence on the model, regardless of the day or night. The red dots (high RH)

are concentrated on the left side (SHAP<0), whereas the blue dots (low RH) are clustered on the right side (SHAP>0), with a

notable separation between the two color groups. This clear distinction signifies a significant negative impact of RH on the

model. Specifically, the red dots, representing higher levels of air humidity, tend to absorb heat through evaporation, thus

mitigating the CUHI phenomenon to a certain degree (Huang et al., 2012; Jiang et al., 2019). Of course, several scholars

contend that humidity can exacerbate heat stress (Mora et al., 2017; Zhang et al., 2023). The CUHII is highly dependent on

variations in WS (Oke et al., 2017; Yang et al., 2020). During the day, WS ranks second in terms of its contribution to

CUHII, but its significance diminishes significantly during the night. Among SWPs, type 4 exhibits the greatest contribution

to CUHII during the day, whereas type 6 dominates during the night. PLAND ranks sixth in contribution during the day, but

gains further importance during the night. Additionally, we observed that the SHAP values for SWPs are more dispersed

during the day, indicating that CUHII was more sensitive to changes in synoptic conditions during this period. Conversely,




the more dispersed SHAP values for human activities during the night suggest that CUHII was more responsive to variations in human activities during this period. We defined the importance of each feature as the mean absolute value of its impact on

the target variable. As depicted in the right subplot (mean SHAP) of Fig. 11, statistical analysis reveals that during the day, the mean SHAP values for SWPs and human activities are 0.12 and 0.10, respectively. During the night, these values change to 0.08 for SWP and 0.16 for human activities. Consequently, during the day, SWPs were more crucial than human activities to influence CUHII, whereas, during the night, human activities surpass SWPs in their importance for CUHII.

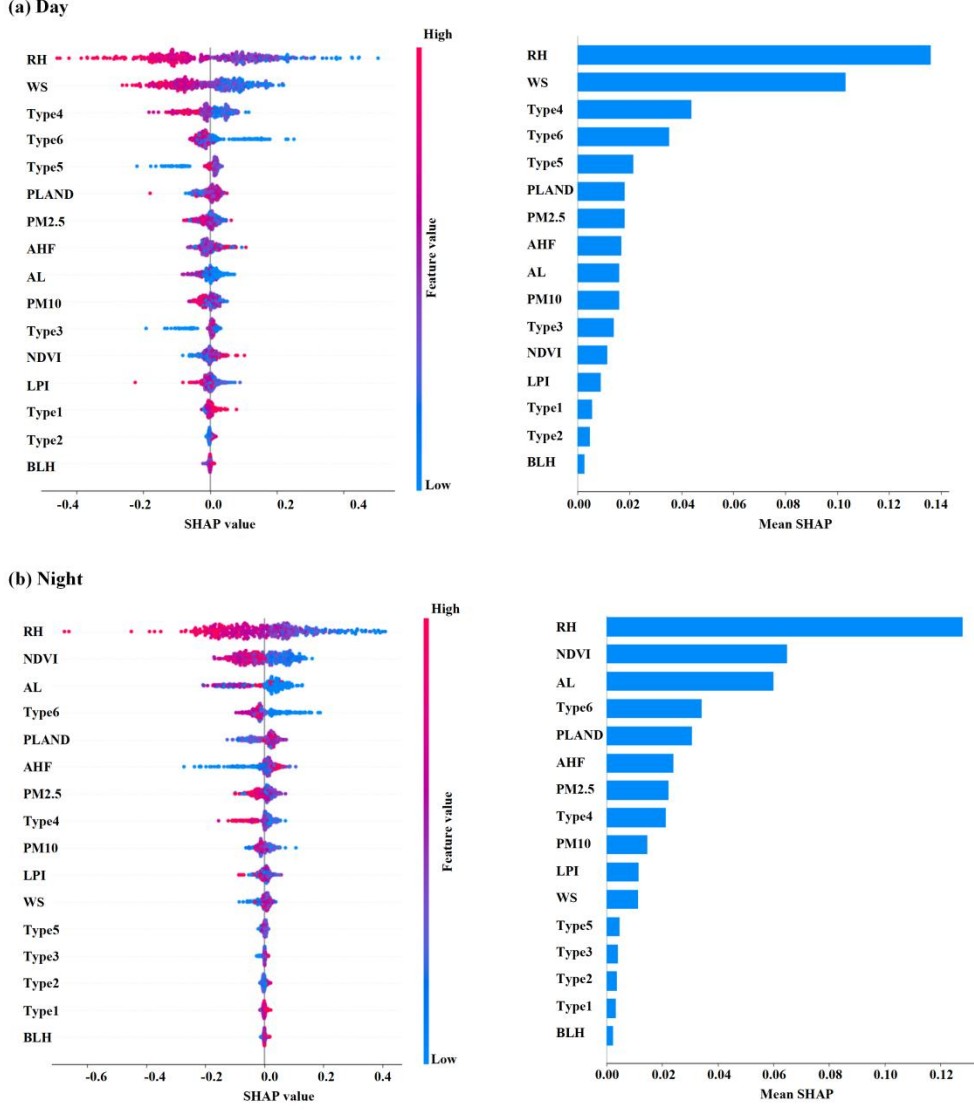

**Figure 11: SHAP plots illustrating the influence factors of day CUHII (a) and night CUHII (b) based on the RF model. The left subplot represents the SHAP values. The right subplot depicts the mean SHAP values, which are used to characterize the importance of various factors.**



In the SHAP plots presented above, the mixed pattern of red and blue dots signifies that the relationships between various factors and CUHII do not adhere to simple positive or negative trends, underscoring the need for an analysis of their nonlinear associations. Within the framework of machine learning, partial dependence plot (PDP) refers to the evaluation of the relationship between a single feature's value and the model's prediction outcomes, while holding all other features constant (Friedman, 2001). Taking type 4 as an illustrative example, this type predominantly occurs in late June, characterized by rainy and overcast days due to the influence of low pressure and moisture transport from the southwestern sea, resulting in high air humidity. As evident from Fig. 12a, during night, PDP gradually decreases with increasing RH, while during the day, the decrease in PDP is more pronounced, indicating that air humidity may exhibit a stronger mitigating effect on CUHII during the day compared to during the night. This explains why type 4's SHAP value ranks third during the day but drops to eighth during the night, as the influence of air humidity on CUHII diminishes. Next, we consider type 5, which typically appears in mid-July, influenced by warm air transported from the southeastern ocean by the subtropical high of the western Pacific, favoring air subsidence, which leads to higher WS. As illustrated in Fig. 12b, during the day, once WS exceeds 1.7 m/s, PDP rapidly decreases, significantly improving the ability to mitigate CUHII. For the entire city, a more consistent wind field at ground level contributes to a stronger heat transport capacity (Xie et al., 2022; Yang et al., 2023). However, during the night, as the WS increases, the PDP remains largely unchanged. This is because urban surfaces undergo radiative cooling during the night, which slows down heat loss. Although WS can facilitate some heat diffusion, its mitigating effect is limited by factors such as dense urban buildings and poor air circulation. Consequently, type 5's SHAP value ranks fifth during day but drops to twelfth during night, as the impact of WS on CUHII rapidly diminishes.

Next, let us examine the partial dependence of CUHII on human activities. During night, when PLAND exceeds 38%, the PDP rapidly increases (Fig. 12c). There may be a threshold for the built-up area, beyond which its contribution to improving CUHII becomes significantly more pronounced. This complex correlation pattern is intimately linked to urban climatic conditions, vegetation coverage within urbanized areas, the frequency of human activities, and seasonal and spatial variations in energy consumption (Guo et al., 2016; Yang et al., 2018; Zhou et al., 2014).). In contrast, during day, the upward trend of PDP is notably weaker than during night. While buildings can intensify CUHII by reducing outgoing longwave radiation and WS, they also block more shortwave solar radiation from reaching the ground, and this shading effect contributes to lowering near-surface air temperatures (Zhang et al., 2016; Krayenhoff & Voogt, 2016; Taleghani et al., 2016; Cai & Xu, 2017). Figure 12d shows that as PM2.5 concentrations increase, PDP gradually decreases. During day, PM2.5 scatters and absorbs part of the solar radiation, reducing the amount of solar radiation reaching the surface and thereby inhibiting the CUHI phenomenon (Yang et al., 2021b). During night, changes in CUHII are more dependent on the energy stored within the urban canopy. The PDP trend exhibits a threshold behavior. When PM2.5 concentrations exceed 40 μg/m³, PM2.5 slows the loss of surface heat and its insulating effect becomes apparent (Li et al., 2020). These results indicate that there is a clear diurnal asymmetry in the modulation of CUHI by SWPs and human activities. This finding provides valuable information on the physical mechanisms of CUHI and the optimization of predictive models.



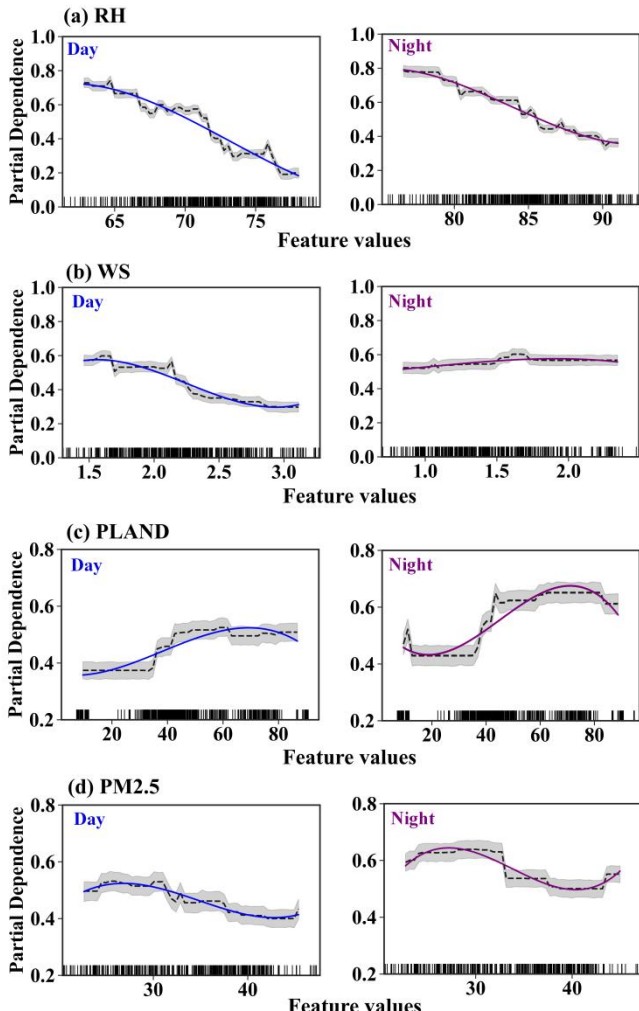

**Figure 12: Partial dependence plots of day CUHII and night CUHII with respect to RH (a), WS (b), PLAND (c), and PM2.5 (d). The blue and red lines represent the fitted curves for day and night, respectively. The gray areas indicate the 95% confidence interval. Rug plots (small vertical lines) along the X axis represent the distribution of the feature values.**

## 4. Discussions

In the context of global climate warming, the frequency and duration of heatwave (HW) events are also increasing worldwide, posing significant challenges to urban thermal environments and resulting in public health issues (IPCC, 2021; Patz et al., 2005; Xu et al., 2016). Next, we will analyze the HW activity patterns under different SWPs in YRDUA.



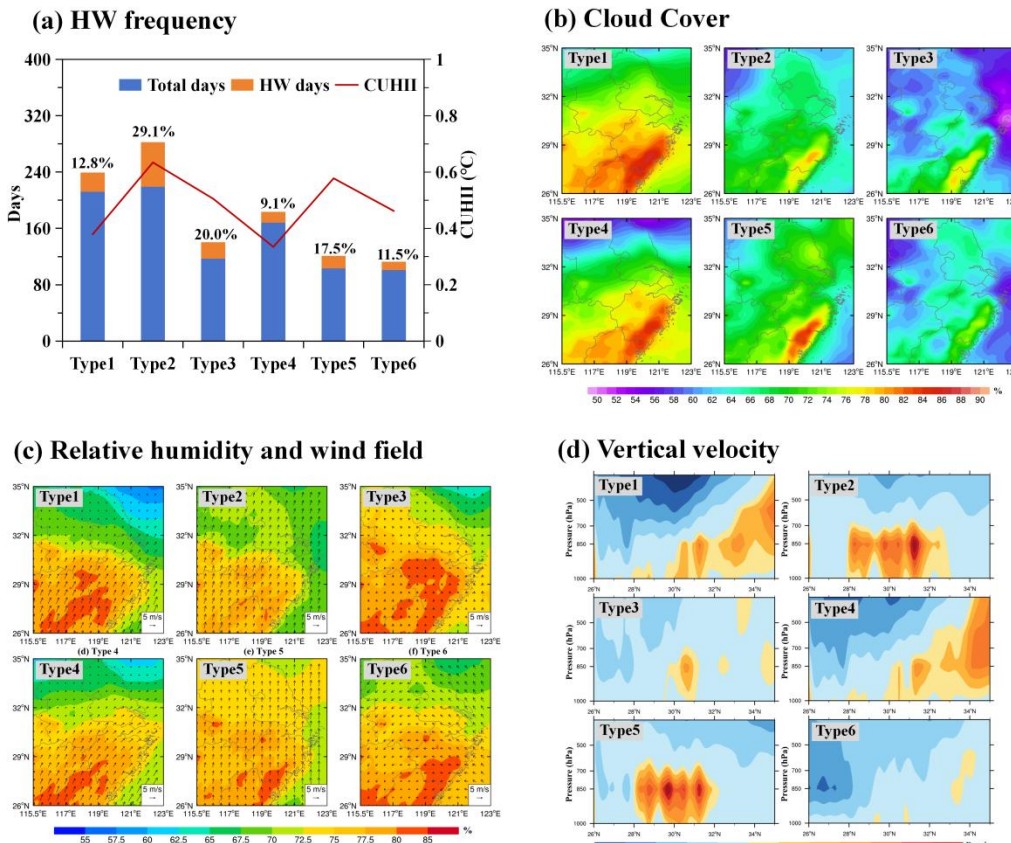

**Figure 13: HW frequency (a), cloud cover (b), relative humidity and wind field (c), and vertical velocity profiles (d) under different SWPs in YRDUA.**

As shown in Fig. 13a, type 2 exhibits the highest frequency of HW events (29.1%), corresponding to the highest CUHII (0.66°C). Type 4, on the other hand, has the lowest HW frequency and the lowest CUHII. The ranking of the HW frequency and CUHII for other synoptic types generally aligns. In Fig. 13b, type 4 has the highest cloud cover, forming a high-value center band in the southeast, while type 2 has a relatively lower cloud cover. Reduced cloud cover improves the reach of solar radiation reaching the surface, contributing to the HW frequency. In Fig. 13c, the relative humidity across the YRDUA is generally high, above 65%. Type 4 displays a high humidity center in the southern part of YRDUA. On the contrary, type 2 exhibits a lower overall relative humidity, influenced by the subtropical high of the western Pacific, which is favorable for the formation of HW events. Fig. 13d presents the zonal profiles of vertical velocity at 500, 700, 850, and 1000 hPa for the six SWPs, with positive values indicating sinking motion and negative values indicating ascending motion. Under type 4, the central and southern regions of YRDUA experience prevalent ascending motion above 850 hPa, where warm air encounters cold air, leading to increased cloud cover and subsequently lower HW frequency. On the contrary, type 2, controlled by the



subtropical high of the western Pacific, exhibits a strong sinking motion in the central region, reducing cloud cover, improving solar radiation reaching the ground, and increasing the HW frequency.

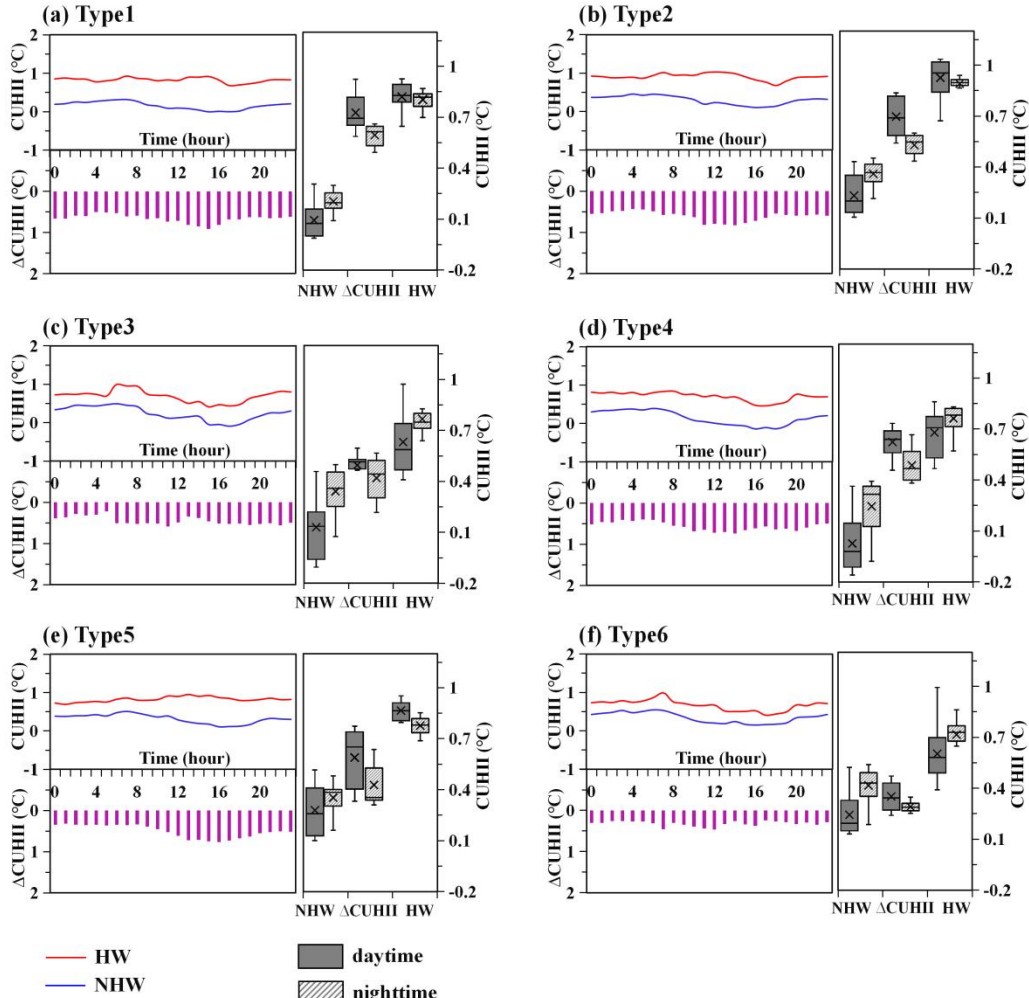

**Figure 14: Synergies between HW and CUHI during HW periods under different SWPs (a-f). In each panel, the upper line chart represents the diurnal variation of CUHII, the lower bar chart represents the diurnal variation of the ΔCUHII during HW periods,**
**and the box plot on the right presents the statistical results for both day and night.**

Previous studies have noted a pronounced amplification of CUHII during HW periods (Li & Bou-Zeid, 2013; Founda et al., 2015; Khan et al., 2020; Ngarambe et al., 2020). Does a similar effect exist for the CUHII in the YRDUA during HW periods? If so, what role does this amplification play in the diurnal cycle of the CUHII? We proceed with our analysis to

explore these questions. Fig. 14 illustrates that the CUHII during HW periods (red line) is significantly higher than that during non-heatwave (NHW) periods (blue line) for all SWPs, indicating a notable amplification of the CUHII in YRDUA during HW periods. Specifically, the differences in CUHII between HW and NHW range from 0.22 ° C to 0.92 ° C (purple



bars), with the most pronounced amplification observed in type 1 and type 2. In particular, amplified CUHII (ΔCUHII) peaks around 15:00 during the day, consistent with previous studies (Tan et al., 2010; Founda et al., 2017), highlighting the crucial

role of day in amplifying CUHII. Statistical analysis of day and night data (box plots) reveals that during NHW periods, CUHII is significantly higher at night than during the day. However, during HW periods, the amplification effect is stronger during the day than at night, significantly narrowing the difference between CUHII at night and day. For example, under type 3, type 4, and type 6, the difference between night and day CUHII decreases by over 35% during HW periods compared to NHW periods. Furthermore, in type 1, type 2, and type 5, during HW periods, day CUHII even surpasses night CUHII. To

gain insight into the underlying physical mechanisms, Fig. 15 compares the diurnal variations of RH and WS between the HW and NHW periods. The results show that, overall, RH during HW periods is generally lower than RH during NHW periods, with the disparity widening significantly during the day. As exemplified by Type 2, the maximum difference in RH coincides with the maximum difference in CUHII at 15:00, suggesting that urban areas during the HW periods are drier than their suburban counterparts, inhibiting the cooling of the evaporation and therefore exacerbating CUHII. Furthermore, WS

analysis indicates that while night WS remains similar between HW and NHW periods, day WS (except for type 6) decreases significantly during HW periods, implying suppressed advective cooling and further contributing to the amplification of the CUHII. In summary, HW events not only significantly amplify CUHII in YRDUA, but also attenuate the diurnal variation of CUHII by modulating local meteorological factors. Given the unique coastal location of YRDUA, the influence of sea-land breeze advection cooling on the diurnal cycle of the CUHII cannot be overlooked. Future research

will focus on typical cities within the region, delving deeper into the effects of sea breezes on the dynamic changes of the urban thermal environment.

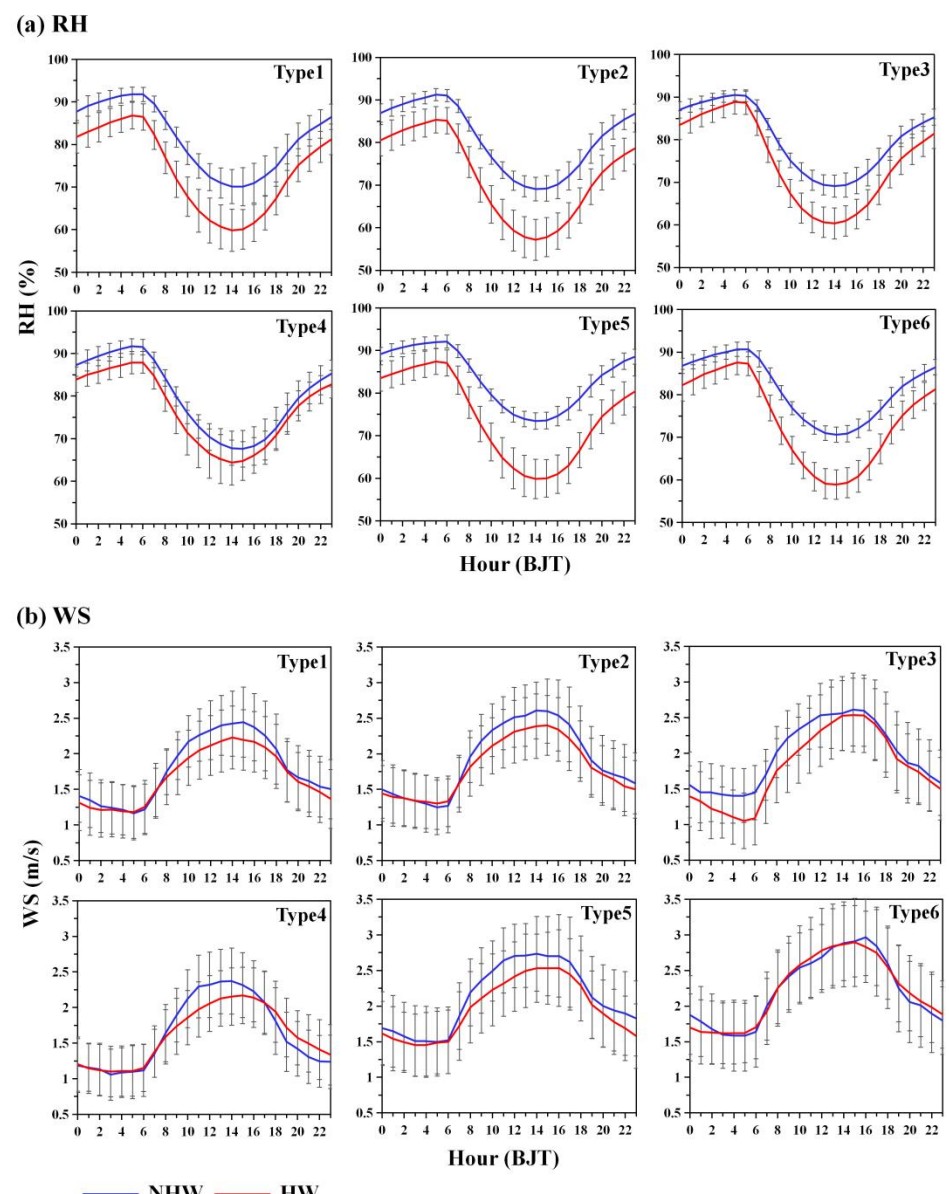

**Figure 15: Diurnal variations of (a) RH and (b) WS during the HW periods (red line) and the NHW periods (blue line).**

## 5 Conclusions


This study systematically analyzed the complex modulation mechanisms of the diurnal cycles of CUHII using objective classification and a machine learning model, taking into account both SWP and human factors. The key findings were summarized as follows.



The CUHII in YRDUA region exhibited a spatial pattern with higher values in the east and lower in the west. night CUHII
was 21.11% stronger than that during day, with June displaying the largest diurnal amplitude. The temporal-spatial dynamics
of the CUHII manifested a pronounced diurnal periodicity. At the synoptic system level, this study clarified the
differentiated impacts of six distinct SWPs on CUHII within the 850 hPa geopotential height field during summer. In
particular, type 2 (dominated by subtropical high pressure) stood out due to its high frequency of occurrence and
accompanying intense CUHII (0.65°C during the day and 0.71 ° C at night). On the contrary, type 4 (jointly influenced by
southwestern moisture and cold air moving southward) was characterized by low frequency and relatively weaker CUHII
values (0.41°C during the day and 0.47 ° C at night). These discoveries indicated that SWPs could play a pivotal role in
regulating the urban heat island effect. Furthermore, this research delved into the contributions of human activities to the
CUHII. Apart from particulate matter, PLAND, LPI, and AHF all exhibited increasing trends over the years, with their
spatial distributions closely mirroring that of the CUHII, again featuring higher values in the east and lower in the west. This
underscored the non-negligible influence of human activities on the CUHII. The RF model further dissected the relative
importance of SWPs and human activities in modulating CUHII, revealing that SWPs dominate during the day, while human
activities become more crucial at night. Further analysis suggested that the diurnal asymmetry in modulation might arise
from the difference in CUHII's partial dependencies on RH, WS, PLAND, and PM2.5 during day and night. Furthermore,
this study evaluated the influence of HW events on the diurnal cycle of CUHII under different SWPs. The results showed
that the amplification of CUHII was significant during HW periods, particularly prominent under type 1 and type 2, with
stronger amplification effects during the day than at night, thus diminishing the diurnal amplitude of the CUHII. This
research not only improved our understanding of the diurnal drivers of CUHII in the YRDUA region, but also provided a
solid scientific basis for formulating targeted urban environmental mitigation strategies.

**Data availability.** Reanalysis data was derived from the fifth generation European Center for Medium Range Weather
Forecasts (https://cds.climate.copernicus.eu/cdsapp#!/home). Meteorological data could be collected from the China
Meteorological Data Service Center (http://data.cma.cn/en). Land cover data are available at
https://zenodo.org/record/5816591 (Yang & Huang, 2021). The AHF data were derived from the inversion of the National
Oceanic and Atmospheric Administration (http://ngdc.NOAA.gov/eog/dmsp/downloadV4composites.html). PM2.5 and
PM10 concentration dataPM10 concentration data could be accessed from the following links:
https://doi.org/10.5281/zenodo.5652265 (Bai et al., 2021a) and https://doi.org/10.5281/zenodo.5652263 (Bai et al., 2021b),
respectively.



**Author contributions.** Tao, S., Yuanjian, Y. conceptualized the study. Tao, S. wrote the original manuscript and plotted all the figures. Yuanjian, Y., Ping, Q., and Simone, L. assisted in the conceptualization and model development. All the authors contributed to the manuscript preparation, discussion, and writing.

**Financial support.** This study was supported by the National Natural Science Foundation of China (42105147), the Joint Research Project for Meteorological Capacity Improvement (22NLTSQ013), and the Collaborative Innovation Fund of the Education Department of Anhui Province (GXXT-2023-050).

**Competing interests.** The contact author has declared that none of the authors has any competing interests.

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
