# Peer review of "The Modulation of Synoptic Weather Patterns and Human Activities on the Diurnal Cycle of Summertime Canopy Urban Heat Island in Yangtze River Delta Urban Agglomeration, China"

_EGUsphere, 2024_

## Author Comment (AC1)

**Response to Review Comments**

Dear Reviewer and Editors:

We are sincerely grateful to the editor and reviewer for their valuable time for reviewing our manuscript. The comments are very helpful and valuable, and we have addressed the issues raised by the reviewer in the revised manuscript. Please find our point-by-point response (in blue text) to the comments (in black text) raised by the reviewer. We have revised the paper according to your comments (highlighted in red text of the revised manuscript).

Sincerely yours,

Dr. Yuanjian Yang, representing all co-authors

**Reviewer #2:**

**This study presents a comprehensive and detailed analysis of the diurnal drivers of the Canopy Urban Heat Island Intensity (CUHII) in the Yangtze River Delta Urban Agglomeration. The manuscript utilizes multiple datasets and methods, making it a robust piece of research. However, the manuscript appears a bit lengthy and the novelty of the findings is not clearly articulated.**

_**Response:**_ Thanks very much for taking time to provide us with such valuable comments that significantly improve the quality of our manuscript. In line with your comments and suggestions, we have revised our manuscript carefully and prepared a list of point-by-point responses below.

Firstly, I have accordingly refined both the abstract and conclusion sections of our manuscript. Indeed, our conclusions are built upon the foundation of existing knowledge. However, as you pointed out, we have employed more advanced weather classification and data mining techniques, which have enabled us to gain a more nuanced understanding of the formation mechanisms of the diurnal cycle of CUHI.

For instance, we have quantified the contributions of SWPs and human activities to the day CUHI and night CUHI, adding depth to the existing literature. Furthermore, our study has uncovered a diurnal asymmetry in the modulation of SWPs and human activities on CUHI, resulting in a significant reduction in the daily amplitude of CUHI. This finding provides a novel perspective for investigating the diurnal cycle and formation mechanisms of the CUHI.

Secondly, I have revised the introduction accordingly, with a particular focus on highlighting the lack of sufficient attention in existing research regarding the combination of SWPs and human activities on the modulation of diurnal cycle of CUHI. Specifically, I have emphasized the gap in understanding the differences in the regulation of daytime and nighttime CUHI by these factors.

Thirdly, thank you very much for your valuable suggestion on streamlining the manuscript. I fully agree that the content prior to section 3.3 was somewhat lengthy, which may have blurred the focus of the article. In response, I have adjusted the overall structure by condensing some discussions and analyses, and have moved some figures to the appendix to enhance the clarity of the paper's logic.

Thank you once again for your valuable feedback, which has greatly improved the quality of our manuscript.

**Major comments:**

**1. In the methodology section, I suggest including a workflow that outlines the datasets and methods used in the study.**

*Response:* Thank you very much for your valuable suggestion. In response to your feedback, a workflow diagram outlining the datasets and methods used in this study has been included.

[Figure]

**Figure R1: The workflow of the datasets and methods used in this paper.**

**2. Figure 1b: It is unclear whether the reference station shown in Figure 1b is the same as the one mentioned in Section 2.3.1. Could you please clarify this in the manuscript?**

*Response:* Thank you for your valuable feedback on Figure 1b. Upon careful review of the station information as per your suggestion, I have identified and corrected a typographical error in Section 2.3.1. Specifically, the number of USs and RSs mentioned in the section were incorrectly stated, and I have now corrected them to 46 and 25, respectively. I sincerely apologize for this mistake and have thoroughly proofread the text to ensure that such errors do not recur.

Furthermore, to clarify any potential confusion regarding the stations depicted in Figure 1b and mentioned in Section 2.3.1, I have attached Table R1, which provides comprehensive information on all selected USs and RSs in the YRDUA region. This table includes station names, station numbers, provinces, longitudes, and latitudes, allowing for easy identification in Figure 1b.

Thank you again for your time and effort in reviewing my manuscript.

**Tab. R1 The information of USs and RSs in YRDUA**

| Station numbers | Types | Provinces | Station names | Longitudes | Latitudes |
|---|---|---|---|---|---|
| 58236 | US | Anhui | Chuzhou | 118.2500 | 32.3500 |

| | | | | | |
|---|---|---|---|---|---|
| 58238 | US | Jiangsu | Nanjing | 118.9000 | 31.9300 |
| 58241 | US | Jiangsu | Gaoyou | 119.4481 | 32.7919 |
| 58242 | US | Jiangsu | Yizheng | 119.1586 | 32.2997 |
| 58245 | US | Jiangsu | Yangzhou | 119.4200 | 32.4100 |
| 58247 | US | Jiangsu | Yangzhong | 119.7983 | 32.2744 |
| 58250 | US | Jiangsu | Jiangyan | 120.1500 | 32.5200 |
| 58252 | US | Jiangsu | Dantu | 119.4667 | 32.1833 |
| 58254 | US | Jiangsu | Haian | 120.4125 | 32.5486 |
| 58255 | US | Jiangsu | Rugao | 120.5675 | 32.3675 |
| 58257 | US | Jiangsu | Jinjiang | 120.2500 | 31.9800 |
| 58321 | US | Anhui | Hefei | 117.0572 | 31.9556 |
| 58334 | US | Anhui | Wuhu | 118.3700 | 31.3800 |
| 58336 | US | Anhui | Maanshan | 118.5667 | 31.7000 |
| 58343 | US | Jiangsu | Changzhou | 119.9781 | 31.8667 |
| 58349 | US | Jiangsu | Suzhou | 120.5600 | 31.4100 |
| 58351 | US | Jiangsu | Jiangyin | 120.3000 | 31.9000 |
| 58352 | US | Jiangsu | Changshu | 120.7667 | 31.6500 |
| 58354 | US | Jiangsu | Wuxi | 120.3500 | 31.6167 |
| 58356 | US | Jiangsu | Kunshan | 121.0000 | 31.4000 |
| 58359 | US | Jiangsu | Wujiang | 120.6167 | 31.1333 |
| 58361 | US | Shanghai | Minhang | 121.3667 | 31.1000 |
| 58362 | US | Shanghai | Baoshan | 121.4447 | 31.3908 |
| 58365 | US | Shanghai | Jiading | 121.1994 | 31.3806 |
| 58367 | US | Shanghai | Xujiahui | 121.4300 | 31.2000 |
| 58370 | US | Shanghai | Pudong | 121.5300 | 31.2300 |
| 58443 | US | Zhejiang | Changxing | 119.8900 | 31.0200 |
| 58449 | US | Zhejiang | Fuyang | 119.9500 | 30.0500 |
| 58451 | US | Zhejiang | Jiashan | 120.9300 | 30.8300 |
| 58452 | US | Zhejiang | Jiaxing | 120.7667 | 30.7333 |
| 58457 | US | Zhejiang | Hangzhou | 120.1600 | 30.2300 |
| 58459 | US | Zhejiang | Xiaoshan | 120.2800 | 30.1800 |
| 58460 | US | Shanghai | Jinshan | 121.2667 | 30.8167 |
| 58461 | US | Shanghai | Qingpu | 121.1167 | 31.1333 |
| 58462 | US | Shanghai | Songjiang | 121.1758 | 31.0200 |
| 58467 | US | Zhejiang | Cixi | 121.2700 | 30.2000 |
| 58468 | US | Zhejiang | Yuyao | 121.1300 | 30.0200 |
| 58561 | US | Zhejiang | Zhenhai | 121.6000 | 29.9800 |
| 58562 | US | Zhejiang | Yinzhou | 121.5000 | 29.8000 |
| 58665 | US | Zhejiang | Hongjia | 121.4167 | 28.6167 |
| 58203 | US | Anhui | Fuyang | 115.7364 | 32.8775 |
| 58424 | US | Anhui | Anqing | 116.9672 | 30.6231 |
| 58141 | US | Jiangsu | Huaian | 118.9269 | 33.6378 |
| 58027 | US | Jiangsu | Xuzhou | 117.1586 | 34.2872 |
| 58549 | US | Zhejiang | Jinhua | 119.6558 | 29.1128 |

| 58659 | US | Zhejiang | Wenzhou | 120.6578 | 28.0250 |
|---|---|---|---|---|---|
| 58223 | RS | Anhui | Mingguang | 117.9892 | 32.8003 |
| 58340 | RS | Jiangsu | Lishui | 119.0639 | 31.6028 |
| 58107 | RS | Anhui | Linquan | 115.2611 | 32.9106 |
| 58235 | RS | Jiangsu | Liuhe | 118.8472 | 32.3686 |
| 58264 | RS | Jiangsu | Rudong | 121.2206 | 32.3422 |
| 58342 | RS | Jiangsu | Jintan | 119.5406 | 31.7103 |
| 58243 | RS | Jiangsu | Xinghua | 119.8172 | 32.9458 |
| 58337 | RS | Anhui | Fanchang | 118.2153 | 31.0558 |
| 58335 | RS | Anhui | Dangtu | 118.5544 | 31.5531 |
| 58339 | RS | Jiangsu | Gaochun | 118.9039 | 31.3333 |
| 58377 | RS | Jiangsu | Taicang | 121.1075 | 31.5136 |
| 58353 | RS | Jiangsu | Zhangjiagang | 120.5697 | 31.8586 |
| 58455 | RS | Zhejiang | Haining | 120.4919 | 30.4792 |
| 58553 | RS | Zhejiang | Shangyu | 120.8133 | 30.0533 |
| 58541 | RS | Zhejiang | Linan | 119.7522 | 30.2969 |
| 58420 | RS | Anhui | Zongyang | 117.2331 | 30.7125 |
| 58565 | RS | Zhejiang | Fenghua | 121.3869 | 29.6917 |
| 58454 | RS | Zhejiang | Deqing | 119.9839 | 30.5253 |
| 58559 | RS | Zhejiang | Tiantai | 120.9706 | 29.1528 |
| 58320 | RS | Anhui | Feixi | 117.0303 | 31.6081 |
| 58366 | RS | Shanghai | Chongming | 121.4928 | 31.6664 |
| 58038 | RS | Jiangsu | Shuyang | 118.7836 | 34.0911 |
| 58012 | RS | Jiangsu | Fengxian | 116.6561 | 34.6719 |
| 58546 | RS | Zhejiang | Pujiang | 119.8722 | 29.4750 |
| 58751 | RS | Zhejiang | Pingyang | 120.5731 | 27.6686 |

**3. Line 205: The statement, "After 18:00 BJ, as the solar altitude angle decreases, the effective radiation in suburban areas gradually increases, accelerating atmospheric heat loss," is confusing. Typically, as solar altitude decreases, radiation decreases, which should not lead to an increase in effective radiation in suburban areas. Could you please provide a more detailed explanation or consider revising this statement for accuracy?**

*Response:* I apologize for the confusion caused by my unclear statement. Let me provide a more detailed and accurate explanation in line :

"After 18:00 BJT, as the solar altitude angle decreases, the shortwave radiation from the sun correspondingly diminishes. For suburban areas, the net radiation generally turns negative after sunset, leading to a stable atmospheric stratification where the

entire underlying surface is in a state of heat loss, resulting in an increased cooling rate (Zhang et al., 2005; Liu et al., 2013). However, in urban areas, due to the accumulation of more heat, long-wave radiation from the ground continues to supply heat to the atmosphere. The urban underlying surface is characterized by dense construction, leading to much lower Sky View Factor (SVF) in streets compared to suburban areas. Longwave radiation from the ground undergoes multiple reflections between walls and the ground, significantly reducing the amount of heat lost from the surface to the atmosphere (Drach et al., 2018; Tian et al., 2019). In addition, high-rise buildings in urban areas with lower SVF tend to experience lower wind speed (Hang et al., 2011). These factors collectively contribute to a rapid widening of the temperature difference between urban and suburban areas during the night."

**Reference:**

Zhang, J., Meng, Q., Li, X., Yang, L.: Urban Heat Island Variations in Beijing Region in Multi Spatial and Temporal Scales.Scientia Geographica Sinica, 31, 11, 6, https://doi.org/10.13249/j.cnki.sgs.2011.011.1349, 2005.

Liu, W., Yang, P., You, H., Zhang, B.: Heat island effect and diurnal temperature range in Beijing area. Climatic and Environmental Research, 18, 2, 171–177, https://doi.org/10.3878/j.issn.1006-9585.2012.11147, 2013.

Drach, P., Kru¨ger, E. L., Emmanuel, R.: Effects of atmospheric stability and urban morphology on daytime intra-urban temperature variability for Glasgow, UK. Sci. Total Environ., 627, 782–791, https://doi.org/10.1016/j.scitotenv.2018.01.285, 2018.

Tian, Y., Miao, J.: Overview of Mountain-Valley Breeze Studies in China. Meteorological Science and Technology, 47, 1, 11. https://doi.org/10.19517/j.1671-6345.20170777, 2019.

Hang, J., Li, Y., Sandberg, M.: Experimental and numerical studies of flows through and within high-rise building arrays and their link to ventilation strategy. J Wind Eng Ind Aerodyn, 99, 1036–1055, https://doi.org/10.1016/j.envsoft.2016.06.021, 2011.

**4. Line 210: The manuscript states, "Before sunrise, between 0:00 and 7:00, the cooling rates of urban and suburban temperatures are similar, causing the CUHII to gradually increase to its daily maximum value of 0.65 °C." However, it is generally understood that urban areas, due to their heat storage in built materials and the canopy structure, would have a slower cooling rate compared to suburban areas. Please provide reasons that support this claim of similar cooling rates.**

*Response:* I apologize for the error in the manuscript. You are correct in pointing out that typically, urban areas experience slower cooling rates compared to suburban areas due to their heat storage in built materials and canopy structure. The statement in the manuscript was a mistake and should be revised. The correct description is:

"Compared to urban areas, suburbs can be regarded as cooling sources (Mirzaei & Haghighat, 2010; Yang et al., 2024). Before sunrise, between 0:00 and 7:00, the cooling rate in urban areas consistently remains lower than that in suburban areas, leading to a gradual increase in the CUHII to its daily maximum value of 0.65°C."

**Reference:**

Mirzaei, P. A. & Haghighat, F.: Approaches to study urban heat island—abilities and limitations. Build. Environ., 45, 2192–2201, https://doi.org/10.1016/j.buildenv.2010.04.001, 2010.

Yang, M., Ren, C., Wang, H., Wang, J., Feng, Z., Kumar, P., Haghighat, F., Cao, S.: Mitigating urban heat island through neighboring rural land cover. Nature Cities, 1, 522–532, https://doi.org/10.1038/s44284-024-00091-z, 2024.

**5. What factors contribute to the similarity in CUHII magnitude during the night and daytime in July?**

*Response:* Thank you for your insightful question, which aligns well with our research direction. As shown in Fig. R2, the top three synoptic weather classifications in July are Type 2, Type 5, and Type 1. Previous studies have indeed observed a significant amplification of CUHII during heatwave (HW) periods (Li & Bou-Zeid, 2013; Founda et al., 2015; Khan et al., 2020; Ngarambe et al., 2020).

We have analyzed the diurnal variation of CUHII during HW and non-heatwave (NHW) periods under these three weather conditions. As illustrated in Fig. R3, the CUHII during HW periods (red line) is notably higher than during NHW periods (blue line) for these three synoptic weather patterns. Further analysis reveals that, under these three weather conditions, the amplified CUHII (ΔCUHII) during the daytime exceeds that at night, aligning with previous studies (Tan et al., 2010; Founda et al., 2017). This highlights the crucial role of daytime in amplifying CUHII. Consequently, the diurnal asymmetry in CUHII amplification due to heatwaves results in daytime CUHII surpassing nighttime CUHII in July.

In the future, we plan to further investigate the physical mechanisms behind this phenomenon. Thank you again for your valuable input.

[Figure]

**Figure R2: (a) Daily, (b) Interannual and (c) Monthly occurrence frequencies of the six SWPs in YRDUA from 2011 to 2020.**

[Figure]

**Figure R3: Synergies between HW and CUHI during HW periods under Type1 (a), Type2 (b), and Type5 (c). In each panel, the upper line chart represents the diurnal variation of CUHII, the lower bar chart represents the diurnal variation of the ΔCUHII during HW periods, and the box plot on the right presents the statistical results for both day and night.**

**Reference:**

Li, D., Bou-Zeid, E.: Synergistic Interactions between Urban Heat Islands and Heat Waves: The Impact in Cities Is Larger than the Sum of Its Parts. Journal of Applied Meteorology and Climatology, 52,9, 2051-2064, https://doi.org/10.1175/JAMC-D-13-02.1, 2013.

Founda, D., Pierros, F., Petrakis, M., Zerefos, C.: Interdecadal variations and trends of the urban heat island in Athens (Greece) and its response to heat waves. Atmospheric Research,161, 1–13, https://doi.org/10.1016/j.atmosres.2015.03.016, 2015.

Khan, H. S., Paolini, R., Santamouris, M., Caccetta, P.: Exploring the synergies between urban overheating and heatwaves (HWs) in Western Sydney, Energies, 13, 2, 470, https://doi.org/10.3390/en13020470, 2020.

Ngarambe, J., Nganyiyimana, J., Kim, I., Santamouris, M., Yun, G. Y.: Synergies between urban heat island and heat waves in Seoul: The role of wind speed and land use characteristics. PLoS ONE, 15, 12, https://doi.org/10.1371/journal.pone.0243571, 2020.

Tan, J., Zheng, Y., Tang, X., Guo, C., Li, L., Song, G., Zhen, X., Yuan, D., Kalkstein, A.J., Li, F., Chen, H.: The urban heat island and its impact on heat waves and

human health in Shanghai. International Journal of Biometeorology, 54, 75–84, https://doi.org/10.1007/s00484-009-0256-x, 2010.

Founda, D., & Santamouris, M.: Synergies between Urban Heat Island and Heat Waves in Athens (Greece), during an extremely hot summer (2012). Scientific Reports, 7, https://doi.org/10.1038/s41598-017-11407-6, 2017.

**6. Line 425: Are there any potential reasons or mechanisms that could explain the significant decrease in daytime WS during Heat Wave?**

*Response:* Thank you for your meticulous observation. Indeed, the significant decrease in daytime wind speed (WS) during heatwaves compared to non-heatwave periods is a notable phenomenon that I had not previously focused on. Upon reviewing other scholars' research (Ao et al., 2019; Shu et al., 2023), I found similar results, as illustrated in Fig. R4 and Fig. R5. However, these studies did not delve deeply into this particular phenomenon.

As we know, heatwaves are often associated with high-pressure systems. Under such systems, the prevailing sinking air currents suppress vertical air motion, reduce cloud cover, and allow the ground to receive more sunlight, leading to increased temperature gradients and enhanced atmospheric stability, which contribute to decreased wind speeds during heatwaves (Gao et al., 2023; Ji et al., 2024).

Regarding why the decrease in daytime wind speed is more pronounced during heatwaves, I speculate that during the daytime, with the intensification of solar radiation, the stability of the high-pressure system further increases. This slows down the rate of air sinking and flowing towards low-pressure areas. Additionally, during the daytime, solar radiation causes the ground temperature to rise more rapidly, further reducing wind speeds. In summary, the more significant decrease in daytime wind speed during heatwaves compared to non-heatwave periods may be attributed to the combined effects of changes in temperature gradients, the influence of high-pressure systems, and enhanced solar radiation. In the future, we will further explore the underlying physical mechanisms.

Thank you again for your valuable insights.

[Figure]

**Figure R4: Diurnal variation curve of mean 10-meter wind speed at Shanghai during heatwaves and non-heatwave periods. (Ao, et al., 2019).**

[Figure]

**Figure R5: Diurnal profiles of the spatial-averaged (a-d) 10-m wind speed (Ws) at the urban and rural/cropland areas, and(e-h) 10-m wind sped difference(Ws) between the urban and rural/cropland areas. The shaded area in grey indicates the night-time hours. (Shu, et al., 2023).**

**Reference:**

Ao, X., Tan, J., Zhi, X., Guo, J., Lu, Y., Liu, D.: Synergistic interaction between urban heat island and heat waves and its impact factors in Shanghai. Acta Geographica Sinica, 74, 9, 1789–1802, https://doi.org/10.11821/dlxb201909007, 2019.

Gao, Y., Shen, X., Dong, W., Zhaoo, L., Luo, Y., Wang, Y.: The synergy of urbanization and western Pacific subtropical high intensification on compound

heat waves in China. Transactions of Atmospheric Sciences, 46, 1, 119–131, https://doi.org/10.13878/j.cnki.dqkxxb.20210311001, 2023.

Ji, X., Chen, G., Chen, J., Xu, L., Lin, Z., Zhang, K., Fan, X., Li, M., Zhang, F., Wang, H., Huang, Z., Hong, Y.: Meteorological impacts on the unexpected ozone pollution in coastal cities of China during the unprecedented hot summer of 2022. The Science of the Total Environment, 914, 170035, https://doi.org/10.1016/j.scitotenv.2024.170035, 2024.

Shu, C., Gaur, A., Wang, L., Lacasse, M.A.: Evolution of the local climate in Montreal and Ottawa before, during and after a heatwave and the effects on urban heat islands. Science of the Total Environment 890, 164497, https://doi.org/10.1016/j.scitotenv.2023.164497, 2023.

**7. While the study provides valuable insights into the modulation of CUHII by SWPs and human activities in the YRDUA, it would be beneficial to discuss whether these findings can be extrapolated to other regions.**

*Response:* Thank you for your constructive comments. The current study focuses on the YRDUA with a shared climatic background, and investigates the diurnal patterns of the CUHII from the perspectives of SWPs and human activities. Indeed, due to variations in climate, topographic circulation, urban morphology, and patterns of human activities, the specific mechanisms and factors influencing the CUHII may differ across regions.

In the future, we will utilize the methods and framework employed in this study as a useful starting point for research in other regions. By taking into account local-specific environmental factors, we aim to continue exploring the modulation of CUHII by SWP and human activities.

Thank you once again for your valuable insights.

**8. The language needs improvement.**

*Response:* Thank you for your suggestion. I have thoroughly checked the spelling and grammar throughout the manuscript, enhancing the overall readability of the text. I

appreciate your feedback and have made the necessary improvements to address your concerns.

**Minor comments:**

**1. Line 71: Please provide the full name of the WPSH at its first mention.**

*Response:* Apologies for the oversight. I have now included the full name of the Western Pacific Subtropical High (WPSH) at its first mention and checked the entire manuscript to prevent similar errors from occurring.

---

## Author Comment (AC2)

**Response to Review Comments**

Dear Reviewer and Editors:

We are sincerely grateful to the editor and reviewer for their valuable time for reviewing our manuscript. The comments are very helpful and valuable, and we have addressed the issues raised by the reviewer in the revised manuscript. Please find our point-by-point response (in blue text) to the comments (in black text) raised by the reviewer. We have revised the paper according to your comments (highlighted in red text of the revised manuscript).

Sincerely yours,

Dr. Yuanjian Yang, representing all co-authors

**Reviewer #1:**

**The study focuses on the contribution of the canyon urban heat island intensity in the daytime and nighttime by analysing different datasets. The analysis is comprehensive and the whole story is also very organised. However, I have one major comment which suggests the author address.**

**Until now, studies have focused a lot on the mechanism of the canyon UHI, especially the intensity. Many previous studies have also focused on the reason for UHI. The study has two main conclusions: 1) CUHI is larger during nighttime and under the high-pressure system; 2) synoptic weather patterns have a more pronounced influence on day CUHII, but human activities dominated night CUHII. These two points are not new findings; they can be easily found and learnt from the previous literature and even textbooks. Thus, what is the significant contribution of the current work? Indeed, authors applied more advanced and updated analysis methods, yet what are the new findings, which are similar to the previous or different to the previous?**

**I would also suggest the authors reconstruct the abstract and introduction. The current version of the abstract cannot fully reflect significance. In the introduction, authors should highlight the combination of the synoptic and human activities! Similarly, more discussion and explanation should focus on section 3.3 in the results. The analysis and results in the previous sections are a bit lengthy, which makes the focus of the article not sharp enough.**

*Response:* Thanks very much for taking time to provide us with such valuable comments that significantly improve the quality of our manuscript. In line with your comments and suggestions, we have revised our manuscript carefully and prepared a list of point-by-point responses below.

Firstly, I have accordingly refined both the abstract and conclusion sections of our manuscript. Indeed, our conclusions are built upon the foundation of existing knowledge. However, as you pointed out, we have employed more advanced weather classification and data mining techniques, which have enabled us to gain a more nuanced understanding of the formation mechanisms of the diurnal cycle of CUHI. For instance, we have quantified the contributions of SWPs and human activities to the day CUHI and night CUHI, adding depth to the existing literature. Furthermore, our study has uncovered a diurnal asymmetry in the modulation of SWPs and human activities on CUHI, resulting in a significant reduction in the daily amplitude of CUHI. This finding provides a novel perspective for investigating the diurnal cycle and formation mechanisms of the CUHI.

Secondly, I have revised the introduction accordingly, with a particular focus on highlighting the lack of sufficient attention in existing research regarding the combination of SWPs and human activities on the modulation of diurnal cycle of CUHI. Specifically, I have emphasized the gap in understanding the differences in the regulation of daytime and nighttime CUHI by these factors.

Thirdly, thank you very much for your valuable suggestion on streamlining the manuscript. I fully agree that the content prior to section 3.3 was somewhat lengthy, which may have blurred the focus of the article. In response, I have adjusted the overall structure by condensing some discussions and analyses, and have moved some

figures to the appendix to enhance the clarity of the paper's logic.

Lastly, to enhance the organization and facilitate the reviewer's understanding of the manuscript, I have attached a workflow diagram in my response, outlining the datasets and methods utilized in this study.

Thank you once again for your valuable feedback, which has greatly improved the quality of our manuscript.

[Figure]

**Figure R1: The workflow of the datasets and methods used in this paper.**

**Minor comments:**

**1. Line 154, reference format typo.**

***Response:*** According to your comments, the reference format typo has been corrected.

I have carefully addressed each of your minor comments and double-checked the entire manuscript for any other potential issues.

**2. Please also indicate the data period and temporal resolution of the ERA5 dataset.**

***Response:*** I apologize for the lack of clarity in my previous submission.

To clarify, the data period for the specific subset of the ERA5 dataset used in our study spans the months of June to August from 2011 to 2020. The temporal resolution

of the dataset is hourly, providing a detailed and comprehensive view of weather and climate conditions over this time frame.

**3. Section 2.3.1: More explanation of the calculation of CUHII. There are 43 USs and 27 RSs, for each US, which RS is selected to be linked with to get the CUHII?**

*__Response:__* Thank you for bringing this clarification to our attention. The method used to calculate CUHII was specifically based on comparing the air temperature differences between USs and RSs during the summertime (Ren et al., 2007; Yang et al., 2022).

$$CUHII = T_{USs} - T_{RSs} \qquad (1)$$

In above equation, CUHII is the canopy urban heat island intensity during the summertime, $T_{USs}$ is the air temperature of the USs, and $T_{RSs}$ is the summer air temperature of the RSs (Ren et al., 2007; Yang et al., 2022).

In addition, I have attached the information of all selected USs and RSs in the YRDUA region as Table R1, including station names, station numbers, provinces, longitudes, and latitudes. Additionally, I have noticed and corrected the typographical error regarding the number of USs and RSs, which are actually 46 and 25. I apologize for the mistake and have double-checked the text to prevent such errors from occurring again.

**Tab. R1 The information of USs and RSs in YRDUA**

| Station numbers | Types | Provinces | Station names | Longitudes | Latitudes |
|---|---|---|---|---|---|
| 58236 | US | Anhui | Chuzhou | 118.2500 | 32.3500 |
| 58238 | US | Jiangsu | Nanjing | 118.9000 | 31.9300 |
| 58241 | US | Jiangsu | Gaoyou | 119.4481 | 32.7919 |
| 58242 | US | Jiangsu | Yizheng | 119.1586 | 32.2997 |
| 58245 | US | Jiangsu | Yangzhou | 119.4200 | 32.4100 |
| 58247 | US | Jiangsu | Yangzhong | 119.7983 | 32.2744 |
| 58250 | US | Jiangsu | Jiangyan | 120.1500 | 32.5200 |
| 58252 | US | Jiangsu | Dantu | 119.4667 | 32.1833 |
| 58254 | US | Jiangsu | Haian | 120.4125 | 32.5486 |
| 58255 | US | Jiangsu | Rugao | 120.5675 | 32.3675 |
| 58257 | US | Jiangsu | Jinjiang | 120.2500 | 31.9800 |

| 58321 | US | Anhui | Hefei | 117.0572 | 31.9556 |
|---|---|---|---|---|---|
| 58334 | US | Anhui | Wuhu | 118.3700 | 31.3800 |
| 58336 | US | Anhui | Maanshan | 118.5667 | 31.7000 |
| 58343 | US | Jiangsu | Changzhou | 119.9781 | 31.8667 |
| 58349 | US | Jiangsu | Suzhou | 120.5600 | 31.4100 |
| 58351 | US | Jiangsu | Jiangyin | 120.3000 | 31.9000 |
| 58352 | US | Jiangsu | Changshu | 120.7667 | 31.6500 |
| 58354 | US | Jiangsu | Wuxi | 120.3500 | 31.6167 |
| 58356 | US | Jiangsu | Kunshan | 121.0000 | 31.4000 |
| 58359 | US | Jiangsu | Wujiang | 120.6167 | 31.1333 |
| 58361 | US | Shanghai | Minhang | 121.3667 | 31.1000 |
| 58362 | US | Shanghai | Baoshan | 121.4447 | 31.3908 |
| 58365 | US | Shanghai | Jiading | 121.1994 | 31.3806 |
| 58367 | US | Shanghai | Xujiahui | 121.4300 | 31.2000 |
| 58370 | US | Shanghai | Pudong | 121.5300 | 31.2300 |
| 58443 | US | Zhejiang | Changxing | 119.8900 | 31.0200 |
| 58449 | US | Zhejiang | Fuyang | 119.9500 | 30.0500 |
| 58451 | US | Zhejiang | Jiashan | 120.9300 | 30.8300 |
| 58452 | US | Zhejiang | Jiaxing | 120.7667 | 30.7333 |
| 58457 | US | Zhejiang | Hangzhou | 120.1600 | 30.2300 |
| 58459 | US | Zhejiang | Xiaoshan | 120.2800 | 30.1800 |
| 58460 | US | Shanghai | Jinshan | 121.2667 | 30.8167 |
| 58461 | US | Shanghai | Qingpu | 121.1167 | 31.1333 |
| 58462 | US | Shanghai | Songjiang | 121.1758 | 31.0200 |
| 58467 | US | Zhejiang | Cixi | 121.2700 | 30.2000 |
| 58468 | US | Zhejiang | Yuyao | 121.1300 | 30.0200 |
| 58561 | US | Zhejiang | Zhenhai | 121.6000 | 29.9800 |
| 58562 | US | Zhejiang | Yinzhou | 121.5000 | 29.8000 |
| 58665 | US | Zhejiang | Hongjia | 121.4167 | 28.6167 |
| 58203 | US | Anhui | Fuyang | 115.7364 | 32.8775 |
| 58424 | US | Anhui | Anqing | 116.9672 | 30.6231 |
| 58141 | US | Jiangsu | Huaian | 118.9269 | 33.6378 |
| 58027 | US | Jiangsu | Xuzhou | 117.1586 | 34.2872 |
| 58549 | US | Zhejiang | Jinhua | 119.6558 | 29.1128 |
| 58659 | US | Zhejiang | Wenzhou | 120.6578 | 28.0250 |
| 58223 | RS | Anhui | Mingguang | 117.9892 | 32.8003 |
| 58340 | RS | Jiangsu | Lishui | 119.0639 | 31.6028 |
| 58107 | RS | Anhui | Linquan | 115.2611 | 32.9106 |
| 58235 | RS | Jiangsu | Liuhe | 118.8472 | 32.3686 |
| 58264 | RS | Jiangsu | Rudong | 121.2206 | 32.3422 |
| 58342 | RS | Jiangsu | Jintan | 119.5406 | 31.7103 |
| 58243 | RS | Jiangsu | Xinghua | 119.8172 | 32.9458 |
| 58337 | RS | Anhui | Fanchang | 118.2153 | 31.0558 |
| 58335 | RS | Anhui | Dangtu | 118.5544 | 31.5531 |

| | | | | | |
|---|---|---|---|---|---|
| 58339 | RS | Jiangsu | Gaochun | 118.9039 | 31.3333 |
| 58377 | RS | Jiangsu | Taicang | 121.1075 | 31.5136 |
| 58353 | RS | Jiangsu | Zhangjiagang | 120.5697 | 31.8586 |
| 58455 | RS | Zhejiang | Haining | 120.4919 | 30.4792 |
| 58553 | RS | Zhejiang | Shangyu | 120.8133 | 30.0533 |
| 58541 | RS | Zhejiang | Linan | 119.7522 | 30.2969 |
| 58420 | RS | Anhui | Zongyang | 117.2331 | 30.7125 |
| 58565 | RS | Zhejiang | Fenghua | 121.3869 | 29.6917 |
| 58454 | RS | Zhejiang | Deqing | 119.9839 | 30.5253 |
| 58559 | RS | Zhejiang | Tiantai | 120.9706 | 29.1528 |
| 58320 | RS | Anhui | Feixi | 117.0303 | 31.6081 |
| 58366 | RS | Shanghai | Chongming | 121.4928 | 31.6664 |
| 58038 | RS | Jiangsu | Shuyang | 118.7836 | 34.0911 |
| 58012 | RS | Jiangsu | Fengxian | 116.6561 | 34.6719 |
| 58546 | RS | Zhejiang | Pujiang | 119.8722 | 29.4750 |
| 58751 | RS | Zhejiang | Pingyang | 120.5731 | 27.6686 |

**Reference:**

Ren, G., Chu, Z., Chen, Z., Ren, Y.: Implications of temporal change in urban heat island intensity observed at Beijing and Wuhan stations. Geophysical Research Letters, 34, 5, https://doi.org/10.1029/2006GL027927, 2007.

Yang, Y., Guo, M., Ren, G., Liu, S., Zong, L., Zhang, Y., et al. Modulation of wintertime canopy urban heat island (CUHI) intensity in Beijing by synoptic weather pattern in planetary boundary layer. Journal of Geophysical Research: Atmospheres, 127, e2021JD035988. https://doi.org/10.1029/2021JD035988, 2022.